# HARDMath2: A Benchmark for Applied Mathematics Built by Students as Part of a Graduate Class

**James V. Roggeveen**,[*] **Erik Y. Wang**,[*]
David Ettel, Will Flintoft, Peter Donets, Lucy S. Nathwani, Nickholas Gutierrez,
Anton Marius Graf, Siddharth Dandavate, Arjun Nageswaran, Raglan Ward,
Ava Williamson, Anne Mykland, Kacper K. Migacz, Yijun Wang, Egemen Bostan,
Duy Thuc Nguyen, Zhe He, Marc L. Descoteaux, Felix Yeung, Shida Liu,
Jorge García Ponce, Luke Zhu, Yuyang Chen, Ekaterina S. Ivshina, Miguel Fernandez,
Minjae Kim, Kennan Gumbs, Matthew Scott Tan, Russell Yang, Mai Hoang,
David Brown, Isabella A. Silveira, Lavon Sykes, Ahmed Roman, William Fredenberg,
Yiming Chen, Lucas Martin, Yixing Tang, Kelly Werker Smith, Hongyu Liao,
Logan G. Wilson, Alexander Dazhen Cai, Andrea Elizabeth Biju, Michael P. Brenner
**School of Engineering and Applied Sciences, Harvard University**

## Abstract

Large language models (LLMs) have shown remarkable progress in mathematical problem-solving, but evaluation has largely focused on problems that have exact analytical solutions or involve formal proofs, often overlooking approximation-based problems ubiquitous in applied science and engineering. To fill this gap, we build on prior work and present **HARDMath2**, a dataset of 211 original problems covering the core topics in an introductory graduate applied math class, including boundary-layer analysis, WKB methods, asymptotic solutions of nonlinear partial differential equations, and the asymptotics of oscillatory integrals. This dataset was designed and verified by the students and instructors of a core graduate applied mathematics course at Harvard. We built the dataset through a novel collaborative environment that challenges students to write and refine difficult problems consistent with the class syllabus, peer-validate solutions, test different models, and automatically check LLM-generated solutions against their own answers and numerical ground truths. Evaluation results show that leading frontier models still struggle with many of the problems in the dataset, highlighting a gap in the mathematical reasoning skills of current LLMs. Importantly, students identified strategies to create increasingly difficult problems by interacting with the models and exploiting common failure modes. This back-and-forth with the models not only resulted in a richer and more challenging benchmark but also led to qualitative improvements in the students' understanding of the course material, which is increasingly important as we enter an age where state-of-the-art language models can solve many challenging problems across a wide domain of fields.

## 1 Introduction

Recent advances in large language models (LLMs) have significantly expanded the frontier of automated mathematical reasoning. While early benchmarks largely focused on elementary arithmetic and symbolic algebra, newer datasets have begun to cover much more challenging material, ranging from Olympiad-style competition problems to graduate-level exams in theoretical mathematics. Indeed, recent benchmarks like Glazer et al. [2024] and Phan et al. [2025] include extremely

---

[*]Equal contribution. Dataset available here. Parsing and evaluation code available here.

39th Conference on Neural Information Processing Systems (NeurIPS 2025) Track on Datasets and Benchmarks.

challenging problems created by research mathematicians and resist saturation even by today's most capable LLMs. However, a critical component of advanced mathematics crucial to applied science and engineering is severely underrepresented. In many real-world contexts, equations that model physical systems, such as nonlinear partial differential equations (PDEs), oscillatory integrals, or multi-scale boundary-layer problems, do not admit exact solutions. Instead, analytical insights can be obtained from a sophisticated toolbox of asymptotic methods, perturbation expansions, and matched approximations. The ability to recognize and leverage these techniques is essential not only for human researchers but increasingly for AI systems intended to assist in scientific discovery. Existing benchmarks fail to capture this domain of reasoning in both scope and difficulty.

**HARDMath2** was created to help address this gap, consisting of 211 problems covering core topics from an introductory graduate course in applied mathematics. They consist of a mixture of original problems written by students enrolled in the course and questions adapted from standard textbooks [Bender and Orszag, 2013]. For the novel problems, students were asked to introduce complexities that would require careful reasoning and additional steps to solve. Each problem was solved by a student and peer-reviewed by other students to ensure the correctness of the solution, which served as the ground-truth against which the LLM-generated solutions were compared. Students revised their problems after seeing how the models responded, introducing additional facets that made the problem more difficult.

This collaborative approach led to a benchmark that is both diverse in content and challenging in form. It includes problems involving perturbation theory, nonlinear ordinary and partial differential equations (ODEs and PDEs), and challenging integrals. However, perhaps the most novel aspect of the dataset is how it was constructed. By interacting with LLMs during the problem-writing and problem-solving process, students simultaneously deepened their understanding of the material and were consequently able to write (and solve) problems that were more mathematically involved than the standard textbook problems posed on homework assignments. The interactive process was facilitated using a novel collaboration and evaluation environment in which students were able to contribute problems and improve on each other's work while being quickly graded based on evaluations from different LLMs. This dual perspective treated LLMs both as tools and as test subjects, pushing the students not only to understand the subtleties of solutions but also to craft and learn how to solve ever harder problems. Our results show that even the most advanced models continue to struggle on many of our problems, as well as highlighting the educational value of building an LLM benchmark as part of a graduate class.

## 2 Related work

Evaluations of mathematical reasoning have rapidly evolved alongside the models' capabilities. Early benchmarks played a crucial role in demonstrating the potential of LLMs in quantitative domains. Prominent examples include MATH [Hendrycks et al., 2021], comprising challenging high school competition-style problems, and GSM8K [Cobbe et al., 2021], which focused on multi-step arithmetic reasoning at the grade school level. However, many of these benchmarks are now saturated, giving rise to a new generation of benchmarks explicitly designed to test the limits of advanced mathematical reasoning. In many cases, these datasets have been curated by expert mathematicians and target difficulty levels comparable to graduate studies or mathematical research.

A notable example is Glazer et al. [2024], which was developed through a collaboration involving over 60 mathematicians. The dataset contains original and unpublished problems spanning a wide range of modern pure mathematics, including number theory, algebraic geometry, category theory, and real analysis, and are touted to require hours or even days of effort from human experts. Another important effort is Humanity's Last Exam (HLE) [Phan et al., 2025], which aims to be a broad-coverage dataset at the frontiers of human knowledge. Unlike Glazer et al. [2024], HLE was entirely crowdsourced from experts online, and consists of over 2,500 problems from a wide range of domains, with mathematics constituting just one topic. Other benchmarks have also targeted graduate-level material by sourcing problems directly from textbooks or qualifying examinations, such as GHOSTS [Frieder et al., 2024] (covering functional analysis, topology, and probability), ARB [Sawada et al., 2023] (consolidated from university mathematics qualifying exam problems), and s1-prob [Muennighoff et al., 2025] (from the probability section of Stanford's PhD qualifying exam in statistics). While they contain challenging problems, these datasets are often limited in size and scalability, and tend to focus on abstract or formal mathematics.

Table 1: Comparison of **HARDMath2** with selected advanced mathematical benchmarks. **HARDMath2** distinctively targets graduate-level applied mathematics requiring approximations and features a student-driven, LLM-interactive problem creation process. Our evaluation method is also unique in that the final formula in the model's output is automatically compared against the ground-truth solution.

| Dataset | Math Focus | Problem Sourcing | Evaluation Method | Size |
|---|---|---|---|---|
| FrontierMath | Formal (Exact Solutions) | Expert Creation | Integer solution comparison | Hundreds |
| GHOSTS | Formal (Proof-Based) | Manual Extraction | Manual human grading | 190 |
| ARB | Formal (Proof-Based) | Manual Extraction | LLM-as-a-Judge | 34 |
| HLE | Broad Coverage (Exact Solutions) | Crowd-Sourced | Multiple-choice comparison | 1k |
| MathArena | Olympiad (Exact Solutions) | Expert Creation | Automated formula parsing | 96 |
| **HARDMath2** | Applied (Exact Solutions) | Student-Generated | Automated formula parsing | 211+ |

## 2.1 The need for applied mathematics

There is still a significant gap in coverage of advanced applied mathematics, where approximation methods allow insights to be gained from mathematical problems that might otherwise be intractable. Previous applied mathematics benchmarks, such as Fan et al. [2025], were limited by lack of diversity in problem types, phrasing templatization, and nonobjective method of evaluation. **HARDMath2** addresses these limitations by introducing original problems that have unique forms using a student-driven method for data generation and an objective evaluation method. Finally, while some recent benchmarks [Feng et al., 2025, Chung et al., 2025] target graduate-level physics problems, models will still struggle with such tasks if they cannot apply the prerequisite mathematical techniques covered in **HARDMath2**. Moreover, if a model can solve advanced problems in physics but fails on the underlying math, it indicates that it is using faulty reasoning.

## 2.2 Innovations in benchmark creation and pedagogical value

Benchmarks are typically constructed using top-down, expert-driven design, as seen in FrontierMath and HLE [Glazer et al., 2024, Hendrycks and Wang, 2024], or via extraction from static sources like textbooks or exams [Frieder et al., 2024, Sawada et al., 2023, Petrov et al., 2025]. In contrast, our approach challenges students to design problems as they go through an applied mathematics course, focusing on problems tied to their current studies. We make use of an interactive environment to give students real-time feedback on the difficulty of their problems, which in turn allows them to iteratively increase the difficulty of their examples. Our methodology pushes students to interact with harder problems than typical for this course while fostering a course environment where an LLM becomes a tool for enriching education.

Finally, our evaluation framework allows objective assessment of the model's solution. Many mathematical benchmarks rely on LLM-based grading to evaluate the solution of a model. While this can capture the model's full solution—including its reasoning process—it introduces noise into the evaluation process, since the "grade" provided by an LLM depends highly upon the model and the prompt being used. Consequently, the LLM-as-a-judge approach to evaluation lacks reliability and objectivity. Other benchmarks use human experts to manually grade solutions according to a rubric. The drawback of this approach is the time required to assess the accuracy of a model and its labor-intensive nature. Our evaluations are conducted using an automatic parser that extract the final symbolic results of the model and ground-truth solutions, and compare them at a given point in the domain to determine whether they are numerically-close. While other benchmarks such as MathArena [Petrov et al., 2025] have also implemented automatic parsing of LaTeX solutions, we believe that our parser is so far the most sophisticated applied to mathematical benchmark grading.

## 3 Dataset and pedagogical framework for problem curation

A distinctive aspect of this dataset is that it was created via the assignments of a university course. Students were tasked with designing at least one original problem each week that could not be solved correctly by Google's Gemini 2.0 Flash model, with an emphasis on creating problems in line with current topics being covered in the course. Students also had to provide a solution for their problems, which were required to be parsable by our custom evaluation framework and accompanied by a Colab

notebook containing numerical verification of their approximate solution. Embedding the creation of the dataset into the core of the course turned homework assignments into a pedagogically rich and collaborative experience. LLMs became a tool to increase student engagement with challenging material, rather than a shortcut for bypassing it. The course culminated with an oral final exam for each student, where they were asked about a problem that they personally submitted to the benchmark. This ensured that the students understood the solutions to the problems they submitted, which were on average far more difficult than traditional homework problems for this course.

## 3.1 Problem submission and verification pipeline

Figure 1 shows the pipeline for dataset creation. First, students write and solve new problems as part of their assigned coursework, and add their problems to a Google Sheet editable by all students in the course. To help facilitate this, the teaching staff gave standardized problem statements (discussed in Appendix A.2.2) that students could use to build their own questions. The students were then encouraged to edit these prompts as necessary to generate challenging questions.

Student submissions were required to be parsable and gradable by our evaluation framework, which is discussed in detail in Section 3.1.1. This also meant that students had to carefully design the problem statement (which was given as the prompt to the LLM) to ensure that their problem clearly indicated a unique solution. One example of early trouble students had to rectify was that simply asking for an expansion without specifying the order leads to ambiguity in what an LLM might produce as a solution. Importantly, given that the student's solutions are the ground-truth for comparison against the LLM-generated solutions, they first went through collaborative verification. Students were required to submit with their problem a Google Colab notebook numerically verifying the accuracy of their approximate analytical solution. In addition to problem creation, students also had to verify each other's problems as part of their course assignment, using both the Colab notebook and whatever additional resources they wanted to use. These peer-reviewers were asked to correct any mistakes that were found either in the solutions or in the prompts to ensure the overall correctness of the dataset.

Figure 2 shows an example of the Google Sheet used for adding problems—which includes the original problem-designer, the verifier, the problem, solution, as well as checks on parsability and results from LLM evaluations—and an example of a validation plot generated by a student's Colab notebook. The final version of the problem and solution had to conform to a LaTeX format compatible with symbolic parsing, detailed in Appendix A.2. However, students could request functionality be added to the parser when needed to increase the difficulty of a particular problem (one example was a student who requested support for incomplete Beta functions).

In addition to our already-available dataset and evaluation code, we plan to publicly release the code to integrate LLM evaluation into Google Sheets, since we believe that this capability will be useful in the creation of future benchmarks.

### 3.1.1 Infrastructure for automating parsing and evaluation

We implemented a custom evaluator in Python that was hosted on an external server and interfaced with a custom Google Sheets plugin. This setup allowed students to write their problems, solutions, and any additional information (such as lists of the expected variables in the solution) directly in a spreadsheet. Students could then feed their problem directly from the spreadsheet to the evaluation server with the push of a button (Figure 2c). The parsing code first cleaned the raw LaTeX output by removing unnecessary formatting information before applying a series of regular expression replacement rules to transform LaTeX into a form that SymPy's `parse_expr` function could interpret [Meurer et al., 2017]. Some example transformations handled by our parser included rewriting expressions such as `\sin^2` to `sin(x)^2` and converting integrals like `\int_1^2 e^{-\frac{3}{x}} dx` into `Integral(e**(-3/x), (x, 1, 2))`. We also included support for special functions that were commonly used in our dataset, such as the Gamma function and incomplete Beta functions.

The parsing code deployed on the Google sheet was the same code used in the evaluation across all of the models. This made evaluation a scalable, automated process but one which was not perfect. We did not want to penalize a model's performance based on the parsability of the answer, which may not be reflective of the model's mathematical capabilities. We therefore excluded from the performance metrics any response from a model that was not parsed. Thus, the metrics are based only on problems the model provided a valid, gradable solution, ensuring that our results are indicative of mathematical

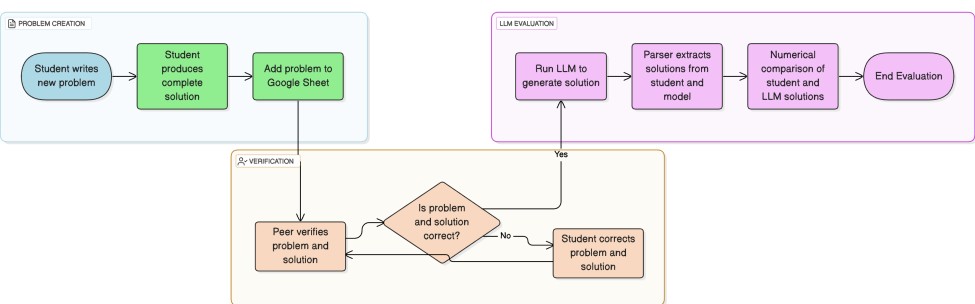

Figure 1: Flowchart of the problem-generation and validation process. Problem creation and validation happen on a collaborative Google Sheet, which includes custom functionality to send problems to a server for LLM querying and evaluation.

reasoning and not ancillary grading-related issues. In practice, this resulted in a very small number of skipped questions for each model.

To compare two SymPy expressions, we numerically evaluated the expression by assigning a random value between one and two to every variable or parameter in the solution. This numerical verification step was necessary because the problems in our dataset may be solved using methods that produce the same formulae but could not be simply compared using SymPy's built-in equality checking. We required that all variables used in the solution be explicitly defined in the prompt and given to the evaluation code as a separate list, which forced problem-setters to write clear and unambiguous problems.

Coupling the parsing and evaluating code to a Google Sheet allowed students to quickly receive feedback from the LLMs on their problems, letting them know if their problem was too easy or too ambiguous when the LLM's solution could not be correctly parsed. This collaborative format also enabled easy peer-verification and let us track how the difficulty of the benchmark evolved in real-time. The semi-automated solution verification and the standardization of prompt formats are discussed more thoroughly in Appendix A.2. The same parsing and evaluation code enabled on our Google Sheet was used to do the final evaluation of the models, as we made only a small subset of LLMs available to the students on the Google sheet. We found that both the problem authors and validators used the Google Sheet's access to the querying and evaluation framework to iteratively increase the difficulty of the problems and to provide a source of sanity-checking by comparing the LLMs reasoning to the student's solution.

### 3.2 Problem types

The problems in **HARDMath2** cover many techniques from the applied mathematician's toolkit, such as the method of dominant balances, optimal truncation, boundary layer analysis, and asymptotic expansions. It goes significantly beyond Fan et al. [2025], which focused on more elementary topics. A major distinction between **HARDMath2** and other mathematical benchmarks is the combination of computational or numerical software, analytical techniques, and "subjective" choices on the part of the problem-solver. For instance, to solve these problems, one must consider different regimes of solution space, the appropriate number of terms to include in approximate expressions, and which approximation method to use. These decisions are be made on a case-by-case basis but involve rigorous mathematical justification, and may be difficult tasks for existing LLMs.

The dataset includes six distinct problem types that leverage these techniques: nonlinear PDEs, nonlinear ODEs, integrals, WKB approximations, boundary layers, and asymptotic series, with a distribution shown in Fig. 3a. As an example, boundary layer theory [Schlichting and Kestin, 1961] is an important applied mathematical tool that rectified apparent contradictions in the theory of aerodynamics. In the 1950s, the theory was further developed [Van Dyke, 1994, Lagerstrom, 2013] leading to both a widely-used toolkit for analyzing physical boundary value problems and a suite of canonically difficult problems that have challenged students for 75 years. A design choice of **HARDMath2** was to focus on problem types that can be made 'harder' to solve (such as by introducing complicated forcing terms, as described in Section 4), rather than simpler problem types

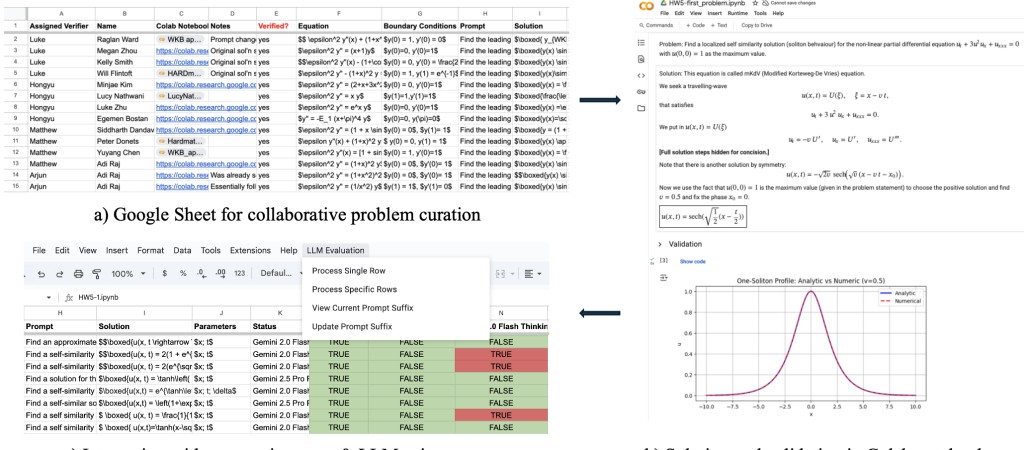

a) Google Sheet for collaborative problem curation

c) Integration with automatic parser & LLM suite

b) Solution and validation in Colab notebook

Figure 2: Problems are collected from students in a Google Sheet, which contains fields for all relevant aspects of the problem and solution, including the prompt passed to the LLM, the regime of interest, and additional parameters. Each student submitted a Colab notebook with their problem demonstrating a numerical comparison of their analytic solution to a full numerical solution, which could then be checked by student verifiers for accuracy. Then, students could instantly run an LLM on their problem (with standardized formatting and solution parsing automatically applied).

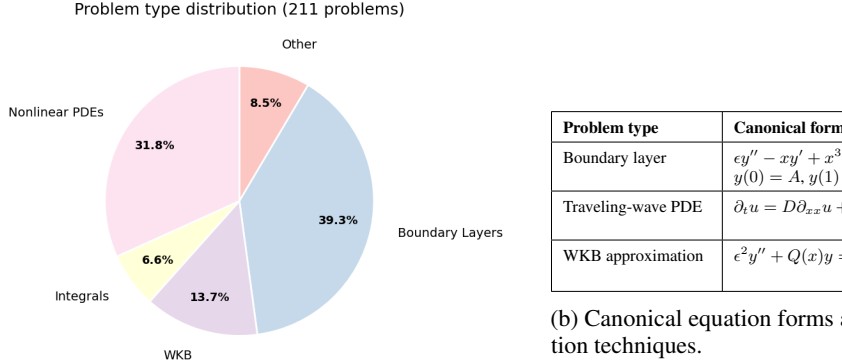

| Problem type | Canonical form | Main tool |
|---|---|---|
| Boundary layer | $\epsilon y'' - xy' + x^3 y = 0$, $y(0) = A, y(1) = B$ | Matched asymptotics |
| Traveling-wave PDE | $\partial_t u = D\partial_{xx} u + R(u)$ | Wave ansatz: $u = f(x - vt)$ |
| WKB approximation | $\epsilon^2 y'' + Q(x)y = 0$ | WKB expansion: $y \sim e^{S/\epsilon}$ |

(b) Canonical equation forms and their primary solution techniques.

(a) Distribution of problem types in **HARDMath2**.

Figure 3: Problem type distribution and associated canonical solution forms in **HARDMath2**.

that have limited complexity. Brief descriptions of the three largest problem classes are described below; the remaining problem types are discussed in detail in Appendix A.1.

### 3.2.1 Boundary layer problems

Boundary-layer problems arise in singularly perturbed differential equations where a small parameter $\epsilon$ multiplies the highest derivative. As $\epsilon \to 0$, solutions typically develop small regions with large gradients to satisfy the boundary conditions. An example of such a problem is given in Figure 3b. In the limit of small $\epsilon$, the leading-order solution is determined by neglecting $\epsilon y''$, yielding $y_{\text{out}}$ but this solution fails to satisfy both boundary conditions. Therefore, one constructs *inner solutions* by rescaling the independent variable near the boundaries to resolve the sharp gradients. The inner and outer solutions are then matched to form a uniformly valid solution.

### 3.2.2 Nonlinear PDEs

Several problems in the benchmark involve nonlinear PDEs, which feature terms where the solution or its derivatives appear nonlinearly; such problems in our benchmark can generally be written as

$$u_t + f(u, u_x, u_{xx}, \ldots) = 0,$$

where $f$ is a nonlinear function of the solution $u$ and its spatial derivatives. These equations can exhibit a range of behaviors depending on which terms in the PDE dominate. Diffusion-dominated solutions spread out over time, advection-dominated equations like Burgers' equation generate shocks and discontinuities, dispersion-dominated systems like the Korteweg-de Vries equation produce solitons and wave trains, and solutions to Laplace's equation yield steady-state patterns or singularities. A given nonlinear PDE may exhibit different behaviors in different regions, depending on which terms dominate. Table 3b shows an example nonlinear PDE for which a traveling-wave ansatz simplifies the equation to an ODE.

### 3.2.3 Wentzel–Kramers–Brillouin (WKB) approximations

We include linear ODEs that are solved with the WKB approximation. The WKB method approximates solutions to linear differential equations with a small parameter $\epsilon$ in the highest derivative term and is particularly effective in regimes where the solution varies rapidly. The solution is assumed to take the form

$$y(x) \sim \exp\left(\frac{1}{\epsilon}S(x)\right),$$

where $S(x)$ is expanded as an asymptotic series in $\epsilon$. Substituting into the differential equation and matching powers of $\epsilon$ yields a sequence of equations: the leading-order term $S_0(x)$ satisfies a Hamilton–Jacobi-type equation, while higher-order terms correct the amplitude. The general solution is typically a linear combination of such exponential modes, matched to boundary conditions.

## 4 Insights from students

To create as challenging of a benchmark as possible, we asked students to design problems that frontier LLMs fail to solve. Using our interactive environment to test candidate problems and solutions, students identified consistent model weaknesses and created new problems that exploited these gaps. Three strategies used by students are as follows.

### 4.1 Structural obfuscation of canonical equations

A common problem-solving technique used by both humans and LLMs involves matching equations to canonical forms. Consider the well-known Fisher–Kolmogorov–Petrovsky–Piskunov (Fisher–KPP) equation:

$$\frac{\partial u}{\partial t} = D \frac{\partial^2 u}{\partial x^2} + r u (1 - u). \tag{1}$$

By introducing an advection term and setting parameters, we can modify the PDE to

$$\frac{\partial u}{\partial t} - \frac{10}{\sqrt{30}} \frac{\partial u}{\partial x} = \frac{2}{5} \frac{\partial^2 u}{\partial x^2} + 2u(1 - u), \tag{2}$$

which is mathematically equivalent to the first form under a Galilean transformation. However, LLMs often fail to recognize the equivalence, as shown below.

> **Error in Gemini 2.5 Pro Analysis**
>
> After transforming the PDE into the traveling wave ODE, the LLM states: "We eliminate the first derivative by choosing specific wave speed and therefore simplify the analysis, we set the coefficient of $f'$ in equation (2) to zero. This eliminates the advection-like term in the ODE and reduces it to a conservative second-order form: $c + \frac{10}{\sqrt{30}} = 0 \quad \Rightarrow \quad c = -\frac{10}{\sqrt{30}}$."
>
> Note: This arbitrary choice of $c$ is incorrect. For the specific ansatz $f(\xi) = \left(1 + e^{a\xi}\right)^{-2}$ that the LLM subsequently (and correctly) proposed for the profile shape, the wave speed $c$ and

the parameter $a$ are co-determined by the PDE's coefficients. This led to an incorrect reduced form and final solution.

## 4.2 Introducing vanishing terms

Differential equations can also be made more LLM-resistant simply by including complex nonlinear terms that evaluate to zero at the specific solution to the problem. For example, the function $u(x,t) = 1 + \tanh(e^{x-2t})$ satisfies the following PDE, which has a nonlinear term crafted to vanish at the solution:

$$\partial_{xx} u = \partial_x u + \frac{(\partial_t u)^2 (1-u)}{2(2u-u^2)} + \underbrace{\frac{((\partial_{xx} u - \partial_x u)(2u - u^2) - \partial_x u\, \partial_t u(u-1))}{((\partial_{tt} u)^2 + u^2)}}_{\text{additional term}}.$$

Importantly, the PDE without the extra term at the end also has $u(x,t)$ as a solution and can often be solved by LLMs, but when confronted with the additional term, they incorrectly guess new functional forms that break their solution.

## 4.3 Initial condition failures

Even when LLMs identify correct general solutions, they often fail to apply initial conditions, which turn a general solution into a specific, physically meaningful one. For example, consider determining the leading-order behavior as $x \to 0^+$ of the third-order ODE:

$$x^4 \frac{d^3 y}{dx^3} = y \tag{3}$$

with initial conditions $y(1) = 1, y'(1) = 0, y''(1) = 0$. A WKB expansion reveals exponential and oscillatory solutions. While models like Gemini 2.0 Flash Thinking and Gemini 2.5 Pro correctly identify this structure, they do not apply the initial conditions to combine the solutions. Even when prompted to do so, they return arbitrary values or keep the symbolic constants. The final answer from Gemini 2.5 Pro after being prompted to solve this equation is shown below:

> **Gemini 2.5 Pro Output for Equation 3**
>
> The leading order behavior of $y(x)$ as $x \to 0^+$ is given by $y(x) \sim Ax^{4/3} \exp(\frac{3}{2x^{1/3}}) \cos(\frac{3\sqrt{3}}{2x^{1/3}} - \phi)$ where $A$ is a positive constant, and $\phi$ is a phase constant. Since the constants $A$ and $\phi$ are fixed by the boundary conditions and are non-zero, this gives the leading behavior.
>
> Final Answer: The final answer is $\boxed{y(x) \sim Ax^{4/3} e^{3/(2x^{1/3})} \cos\left(\frac{3\sqrt{3}}{2x^{1/3}} - \phi\right)}$ for some constants $A > 0$ and $\phi$.
>
> Note: The LLM makes some attempt to narrow down what the constants are, but it does not solve for them completely. It mentions that the initial conditions fix the constants, but it does not list the equations it would use to solve for these constants, nor does it solve them.

# 5 Evaluation

To rigorously assess the abilities of current state-of-the-art language models, we conduct evaluations on a broad set of models. This includes closed-source models like OpenAI's GPT and o-series; Google's Gemini-series; and Anthropic's Sonnet 3.7, as well as open-source models such as Meta Llama 4 and DeepSeek V3. Closed-source models are accessed using their official APIs, while open-source models are tested via Together AI's API.

The results of our evaluation are presented in Tables 2 and 3. While we provide full results with our numerical parser, we also implemented an LLM-as-a-judge with partial credit scoring to better

| Model | Overall | Asymptotic Series | Boundary Layers | Integrals | Nonlinear ODE | Nonlinear PDE | Other | WKB |
|---|---|---|---|---|---|---|---|---|
| Claude 3.7 Sonnet | 11.8 | 33.3 | 4.2 | 61.5 | 28.6 | 1.7 | 16.7 | 24.0 |
| DeepSeek V3 | 18.9 | 0.0 | 22.5 | 46.2 | 42.9 | 6.7 | 0.0 | 18.2 |
| Llama 4 Maverick | 16.7 | 0.0 | 18.3 | 76.9 | 28.6 | 3.3 | 16.7 | 11.5 |
| GPT-4.1 | 7.4 | 0.0 | 1.4 | 38.5 | 16.7 | 5.1 | 0.0 | 14.3 |
| GPT-4o | 4.2 | 0.0 | 0.0 | 23.1 | 28.6 | 2.3 | 0.0 | 6.2 |
| o1 | 45.1 | 0.0 | 47.3 | 71.4 | 50.0 | 31.5 | 50.0 | 55.6 |
| o3 | 52.5 | 50.0 | 64.8 | 76.9 | 40.0 | 35.7 | 16.7 | 53.8 |
| o3-mini | 46.0 | 0.0 | 44.6 | 78.5 | 40.0 | 37.8 | 60.0 | 50.0 |
| o4-mini | 46.1 | 25.0 | 53.6 | 76.9 | 60.0 | 23.1 | 50.0 | 50.0 |
| Gemini 2.0 Flash | 8.5 | 0.0 | 1.3 | 33.3 | 33.3 | 9.3 | 0.0 | 13.6 |
| Gemini 2.5 Flash Thinking | 60.1 | 20.0 | 77.8 | 71.4 | 40.0 | 45.0 | 16.7 | 61.5 |
| Gemini 2.5 Pro Preview | 57.7 | 25.0 | 72.6 | 78.6 | 14.3 | 44.3 | 20.0 | 60.0 |

Table 2: Pass@1 Rates by Model and Question Type

| Model | Boundary Layers | Nonlinear PDE | WKB |
|---|---|---|---|
| DeepSeek V3 | 57.6 | 55.3 | 55.3 |
| o4-mini | 70.2 | 60.9 | 64.7 |
| Gemini 2.5 Pro | 80.5 | 72.3 | 68.7 |

Table 3: LLM-as-a-judge results for boundary layer, nonlinear PDE, and WKB problems.

understand the model's reasoning trajectories and include a subset of these results. Students in the course, in addition to developing the benchmark problems, were asked to solve several of the problems as part of their final oral exam. We have used the detailed rubrics against which these students were scored to provide partial-credit scoring results using an LLM-judge—these rubrics are provided in A.3.

We observe significant disparities across models and problem classes. Among the closed-source models, the Gemini 2.5 family—2.5 Flash Thinking and 2.5 Pro—exhibit the highest accuracies, achieving 60.1% and 57.7% respectively. These results show a marked improvement over prior Gemini models, which only attained 8.5%. This suggests that the "thinking process" built in to the newer Gemini models offer significantly improved mathematical reasoning capabilities. Similar improvements can be seen in OpenAI's o-series, which also demonstrate consistently strong performance. Notably, o3 achieves the highest overall score (52.5%), followed closely by o4-mini (46.1%) and o3-mini (46.0%). In contrast, GPT-4.1 and GPT-4o show substantially worse performance. Even though GPT-4.1 was released relatively recently, its poorer performance may be attributed to its lack of a reasoning mode.

Within the dataset, integrals appear to be the most tractable problem type, with models like Llama 4 Maverick, o1, o3, o3-mini, and Gemini 2.5 Pro Preview all scoring above 70%. This suggests that problems that must conform to a relatively rigid structure may be more easily solved by state-of-the-art models. Nonlinear PDEs, by contrast, are considerably more challenging; only the top-performing models exceed 35% in this category, with o3-mini and Gemini 2.5 variants reaching 37.8% and 44.3–45.0%, respectively. This provides further evidence that problem types which can be more substantially customized—especially via the techniques discussed in Section 4—pose a greater challenge to models.

As stated above, we excluded from the performance metrics any solutions which did not produce parsable solutions, though such occurrences were rare. Every problem in the benchmark had at least one model produce a valid, gradable solution. The exception was DeepSeek R1, which did not follow the instructions to place its solutions in a \boxed{} environment, leading to its exclusion. We found that other low-scoring models (e.g., Claude 3.7 Sonnet) frequently omitted important details from the solution. These examples include using ellipses (...) or unspecific variables to substitute important mathematical details in its reasoning process, and then never substituting the variables or terms back into their final expression.

Other challenges with non-adherence to standards related to some model's tendency to over-format responses. For example, OpenAI's GPT-4o model focused on manipulating display spacing, such as using "{-}" to slightly adjust spacing around operators. These display-only additions needed to be deleted on a case-by-case basis. Further, some models injected unusual Unicode symbols into their responses which needed to be replaced with plain text for parsing. Finally, in a few cases (especially with respect to boundary layer problems) models would return solutions that were not purely analytic expressions but contained nested integrals. While our parser could handle these integrals, sometimes

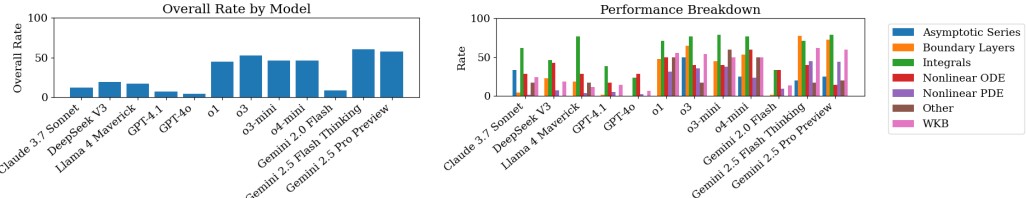

(a) Overall performance of models across all problem types in **HARDMath2**.

(b) Breakdown of model performance across problem types.

Figure 4: Model performance on **HARDMath2**. While (a) shows overall success rates by model, (b) shows significant differences in performance across problem types.

the integrals in the models' responses would not converge numerically and so were skipped after timing out. For more discussion on parsability and instruction-following challenges, see Appendix A.5.

Overall, the results in Figure 4a show that the benchmark remains difficult for even the most capable LLMs currently available. While top-performing models like Gemini 2.5 Pro and o3 demonstrate competence in some categories, no model approaches mastery across all problem types. In categories such as boundary layer theory, nonlinear PDEs, and WKB approximation, the majority of models fail to achieve even moderate accuracy. The consistently low scores outside integrals and basic ODEs highlight persistent gaps in multi-step mathematical reasoning and problem-solving techniques that are fundamental to advanced applied mathematics. These results therefore suggest that while recent advancements in LLMs have advanced the frontier in terms of their mathematical abilities, the specialized mathematical knowledge required by this benchmark still remains largely out of reach.

# 6 Conclusion

We introduce **HARDMath2**, a benchmark that is inspired by previous work [Fan et al., 2025] but contains a variety of harder problem types from a graduate applied mathematics course. The benchmark is unique in both scope and design, comprising 211 original problems across several categories: nonlinear PDEs, integrals, WKB approximations, boundary layer problems, and asymptotic expansions. It expands the current landscape of mathematical benchmarks by introducing more challenging and underrepresented applied mathematics problems involving techniques crucial to real-world scientific and engineering applications. Notably, it also features a novel approach to problem curation that leverages modern LLMs to enhance student learning. We hope that the methodology described for building a benchmark through university courses can be applied to create challenging benchmarks in other quantitative domains in the future. In particular, it is a way for students to challenge themselves beyond the level of a typical homework problem while also generating high-quality problems with limited instructor intervention.

Going forward, we plan to broaden the range of problem types in the benchmark, focusing on the problem types that can be made more difficult. Examples of such problems include asking for more terms in an asymptotic expansion or solving other types of PDEs, like Green's functions. Students similarly found that LLMs themselves could be used to arbitrarily make problems more difficult via prompting, although this would also require more time and effort from the students due to the generation-verification gap [Song et al., 2025]. Finally, our hybrid student and LLM-driven framework for designing original problems that are more challenging than those found in textbooks or assigned on homework assignments can be applied to any quantitative class involving advanced mathematical reasoning, such as courses in physics, statistics, and engineering.

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

# A Appendix

## A.1 Detailed descriptions of problem types

### A.1.1 Boundary layer problems

Many problems in **HARDMath2** are ODEs with boundary layers, where different solutions dominate near a boundary than in the rest of the region (see Section 3.2.1). We provide an example below.

---

**Sample Boundary Layer Problem and Full Solution**

**Problem:** Consider the differential equation:

$$\epsilon y'' - xy + x^3 y = 0$$

Find a uniformly valid approximation to the solution with boundary conditions $y(0) = A$ and $y(1) = B$ in the limit $\epsilon \ll 1$.

---

**Solution:**
Step 1: Find the outer solution Use dominant balance since $\epsilon$ is small:

$$-xy'_{out} + x^3 y_{out} = 0$$

If $x \neq 0$, this simplifies to $y'_{out} = x^2 y_{out}$. This is a separable first-order differential equation:

$$\frac{dy_{out}}{y_{out}} = x^2 dx$$

Integrating both sides:

$$\ln |y_{out}| = \frac{x^3}{3} + C_0$$

$$y_{out}(x) = C e^{x^3/3}$$

where $C$ is an arbitrary constant.
Step 2: Analyze boundary conditions and potential boundary layers.
The outer solution must satisfy one of the boundary conditions. Check $y(0) = A$. $y_{out}(0) = Ce^0 = C$. So, $C = A$. Thus, the outer solution satisfying $y(0) = A$ is:

$$y_{out}(x) = A e^{x^3/3}$$

Check the behavior of this outer solution at $x = 0$ with respect to the original equation.

$$y'_{out}(x) = A x^2 e^{x^3/3}$$

$$y''_{out}(x) = A(2x e^{x^3/3} + x^2 \cdot x^2 e^{x^3/3}) = A(2x + x^4) e^{x^3/3}$$

Substituting these into the full differential equation at $x = 0$:

$$\epsilon y''_{out}(0) - 0 \cdot y'_{out}(0) + 0^3 \cdot y_{out}(0) = \epsilon \cdot 0 - 0 + 0 = 0$$

The outer solution $y_{out}(x) = A e^{x^3/3}$ satisfies the boundary condition at $x = 0$ and is consistent with the full differential equation at $x = 0$. This suggests there is no boundary layer at $x = 0$.
Consider the boundary condition at $x = 1$, $y(1) = B$. The outer solution gives $y_{out}(1) = A e^{1/3}$. If $B \neq A e^{1/3}$ (which is generally the case), the outer solution alone cannot satisfy the boundary condition at $x = 1$. This indicates a boundary layer near $x = 1$.
Step 3: Analyze the boundary layer at $x = 1$.
Let the inner variable be $X = \frac{1-x}{\epsilon^\alpha}$ for some $\alpha > 0$. So $x = 1 - \epsilon^\alpha X$. Thus, the derivatives become:

$$y' = -\frac{1}{\epsilon^\alpha} \frac{dy}{dX}$$

$$y'' = \frac{1}{\epsilon^{2\alpha}} \frac{d^2 y}{dX^2}$$

Substituting these into the differential equation:

$$\epsilon^{1-2\alpha}\frac{d^2y}{dX^2} + \epsilon^{-\alpha}(1 - \epsilon^\alpha X)\frac{dy}{dX} + (1 - 3\epsilon^\alpha X + O(\epsilon^{2\alpha}))y = 0$$

We need to balance the highest derivative term with another dominant term in the boundary layer. The term $\epsilon^{-\alpha}\frac{dy}{dX}$ is likely to be dominant. So, $1 - 2\alpha = -\alpha \implies \alpha = 1$

The stretched variable is $X = \frac{1-x}{\epsilon}$. The equation in terms of $X$ for the inner solution $y_{in,1}(X)$ becomes (substituting $\alpha = 1$):

$$\frac{d^2y_{in,1}}{dX^2} + (1 - \epsilon X)\frac{dy_{in,1}}{dX} + \epsilon(1 - \epsilon X)^3 y_{in,1} = 0$$

To leading order in $\epsilon$, as $\epsilon \to 0$:

$$\frac{d^2y_{in,1}}{dX^2} + \frac{dy_{in,1}}{dX} = 0$$

The general solution is $y_{in,1}(X) = K_1 e^{-X} + K_2$

Step 4: Apply boundary condition and matching for the inner solution at $x = 1$.
The boundary condition at $x = 1$ is $y(1) = B$. In terms of $X$, $x = 1$ corresponds to $X = 0$. So:

$$y_{in,1}(0) = B \implies K_1 e^0 + K_2 = B \implies K_1 + K_2 = B$$

Matching condition: The limit of the inner solution as $X \to \infty$ must match the limit of the outer solution as $x \to 1^-$.

$$\lim_{X\to\infty} y_{in,1}(X) = \lim_{X\to\infty}(K_1 e^{-X} + K_2) = K_2$$

$$\lim_{x\to 1^-} y_{out}(x) = \lim_{x\to 1^-} Ae^{x^3/3} = Ae^{1/3}$$

$$K_2 = Ae^{1/3}$$

Substituting $K_2$ back into $K_1 + K_2 = B$:

$$K_1 + Ae^{1/3} = B \implies K_1 = B - Ae^{1/3}$$

Thus, the inner solution for the boundary layer at $x = 1$ is:

$$y_{in,1}\left(\frac{1-x}{\epsilon}\right) = (B - Ae^{1/3})e^{-(1-x)/\epsilon} + Ae^{1/3}$$

Step 5: Construct the uniformly valid approximation. A common method for constructing a uniformly valid approximation is $y_{unif}(x) = y_{out}(x) + y_{in}(x) - \text{common part}$. So we have:

$$y_{unif}(x) = Ae^{x^3/3} + (B - Ae^{1/3})e^{-(1-x)/\epsilon}$$

Step 6: Verify boundary conditions for the uniform approximation.
At $x = 0$:

$$y_{unif}(0) = Ae^0 + (B - Ae^{1/3})e^{-1/\epsilon} = A + (B - Ae^{1/3})e^{-1/\epsilon}$$

Since $\epsilon \ll 1$, $e^{-1/\epsilon}$ is exponentially small. So $y_{unif}(0) \approx A$.
At $x = 1$:

$$y_{unif}(1) = Ae^{1/3} + (B - Ae^{1/3})e^0 = Ae^{1/3} + B - Ae^{1/3} = B$$

The boundary conditions are satisfied to leading order. The final solution is:

$$\boxed{y(x) \approx Ae^{x^3/3} + (B - Ae^{1/3})e^{-(1-x)/\epsilon}}$$

### A.1.2 Nonlinear PDE problems

We solve many types of nonlinear PDEs in the benchmark (Section 3.2.2). Here, we provide a solution to a PDE dominated by dispersion with a traveling wave solution.

---

**Sample Nonlinear PDE Problem and Full Solution**

**Problem:** Consider the Korteweg-de Vries (KdV) equation:

$$\frac{\partial u}{\partial t} + 6u\frac{\partial u}{\partial x} + \frac{\partial^3 u}{\partial x^3} = 0.$$

Find a soliton solution in the limit $t \Rightarrow \infty$.

---

**Solution:** We seek a traveling wave solution of the form $u(x,t) = f(\xi)$, where $\xi = x - ct$ and $c$ is the constant wave speed. The domain is approximated as $x \in (-\infty, \infty)$ for a localized soliton solution. Substituting the traveling wave ansatz into the KdV equation yields:

$$-cf'(\xi) + 6f(\xi)f'(\xi) + f'''(\xi) = 0$$

Integrating once with respect to $\xi$:

$$f'' + 3f^2 - cf + A = 0$$

For a localized solution, we require $f, f', f'' \to 0$ as $|\xi| \to \infty$. This boundary condition implies the integration constant $A = 0$.

$$f'' + 3f^2 - cf = 0$$

Multiplying by $f'$ to facilitate integration (energy method):

$$f'f'' + 3f^2 f' - cff' = 0$$

This can be written as the derivative of a conserved quantity:

$$\frac{d}{d\xi}\left(\frac{1}{2}(f')^2 + f^3 - \frac{c}{2}f^2\right) = 0$$

Integrating again with respect to $\xi$:

$$\frac{1}{2}(f')^2 + f^3 - \frac{c}{2}f^2 + B = 0$$

Applying the boundary conditions $f, f' \to 0$ as $|\xi| \to \infty$ requires the second integration constant $B = 0$.

$$\frac{1}{2}(f')^2 = \frac{c}{2}f^2 - f^3$$

Rearranging gives:

$$(f')^2 = cf^2 - 2f^3 = f^2(c - 2f)$$

Assuming $f > 0$ within the soliton and taking the square root ($f' = \frac{df}{d\xi}$):

$$\frac{df}{d\xi} = \pm f\sqrt{c - 2f}$$

Separating variables:

$$\frac{df}{f\sqrt{c - 2f}} = \pm d\xi$$

Integrating both sides:

$$\int \frac{df}{f\sqrt{c - 2f}} = \pm \int d\xi = \pm(\xi - \xi_0)$$

where $\xi_0$ is an integration constant representing the initial position.

To evaluate the integral on the left, we use the substitution $f = \frac{c}{2}\text{sech}^2(\theta)$.

$$df = -c\,\text{sech}^2(\theta)\tanh(\theta)\,d\theta$$

The term under the square root becomes:

$$\sqrt{c - 2f} = \sqrt{c - c\,\text{sech}^2(\theta)} = \sqrt{c(1 - \text{sech}^2(\theta))} = \sqrt{c\tanh^2(\theta)} = \sqrt{c}|\tanh(\theta)|$$

Choose the branch where $\tanh(\theta) > 0$:

$$\int \frac{-c\,\text{sech}^2(\theta)\tanh(\theta)\,d\theta}{\left(\frac{c}{2}\text{sech}^2(\theta)\right)\left(\sqrt{c}\tanh(\theta)\right)} = \int \frac{-2}{\sqrt{c}}\,d\theta = -\frac{2}{\sqrt{c}}\theta$$

Equating this to the right side:

$$-\frac{2}{\sqrt{c}}\theta = \pm(\xi - \xi_0)$$

$$\theta = \mp\frac{\sqrt{c}}{2}(\xi - \xi_0)$$

Substituting back into $f = \frac{c}{2}\text{sech}^2(\theta)$:

$$f(\xi) = \frac{c}{2}\text{sech}^2\left(\mp\frac{\sqrt{c}}{2}(\xi - \xi_0)\right)$$

Finally, substituting $\xi = x - ct$, and set $c = 4$ $x_0 = 0$:

$$\boxed{u(x,t) = 2\text{sech}^2(x - 4t)}$$

### A.1.3  WKB approximation problems

We include many ODEs that can be modeled using the WKB approximation; see Section 3.2.3 for an explanation of the technique. We provide a simple example problem below.

---

**Sample WKB Problem and Full Solution with Initial Conditions**

**Problem:** Consider the differential equation:

$$y''(x) = \frac{x}{\epsilon^2}y(x),$$

for small positive $\epsilon$ in the limit $\epsilon \ll 1$, subject to the initial conditions at $x = 1$:

$$y(1) = e^{2/(3\epsilon)}, \quad y'(1) = \frac{1}{\epsilon}e^{2/(3\epsilon)}.$$

---

**Solution:** This equation fits the general WKB form:

$$y'' = R(x)y \quad \text{with} \quad R(x) = \frac{x}{\epsilon^2}.$$

We assume a solution of the form:

$$y(x) \sim \exp\left(\frac{1}{\delta}\sum_{n=0}^{\infty}\delta^n S_n(x)\right).$$

To leading order, we approximate this by truncating after the first two terms:

$$y(x) \sim \exp\left(\frac{1}{\delta}(S_0(x) + \delta S_1(x))\right).$$

We now differentiate using the product rule:

$$y' = \left(\frac{1}{\delta}S'(x)\right)\exp\left(\frac{1}{\delta}S(x)\right), \quad y'' = \left[\left(\frac{1}{\delta}S'(x)\right)^2 + \left(\frac{1}{\delta}S''(x)\right)\right]\exp\left(\frac{1}{\delta}S(x)\right).$$

Substitute into the original differential equation:
$$\left(\frac{1}{\delta}S'(x)\right)^2 + \left(\frac{1}{\delta}S''(x)\right) = \frac{x}{\epsilon^2}.$$

Expanding and collecting powers of $\delta$ gives:
$$\delta^{-2}S_0'^2 + 2\delta^{-1}S_0'S_1' + \delta^{-1}S_0'' + \cdots = \frac{x}{\epsilon^2}.$$

The leading-order balance suggests that $S_0'^2 \sim x$, and we expect $S_0'$ to be large while $S_0''$ remains small. Thus, to leading order, we take:
$$\delta^{-2}S_0'^2 = \frac{x\delta^2}{\epsilon^2}.$$

To match both sides, we must take $\delta = \epsilon$, the small parameter. Substituting back in:
$$S_0'(x)^2 = x \quad \Rightarrow \quad S_0(x) = \pm\int_0^x \sqrt{t}\,dt = \pm\frac{2}{3}x^{3/2}.$$

Now solve for the first-order correction $S_1(x)$. From the remaining terms:
$$2S_0'S_1' + S_0'' = 0.$$

Using $S_0' = \sqrt{x}$ and $S_0'' = \frac{1}{2\sqrt{x}}$, we get:
$$2\sqrt{x}S_1' + \frac{1}{2\sqrt{x}} = 0 \quad \Rightarrow \quad S_1'(x) = -\frac{1}{4x}, \quad S_1(x) = -\frac{1}{4}\ln x.$$

Combining these, we find the two independent asymptotic solutions:
$$y_1(x) \sim x^{-1/4}\exp\left(\frac{2x^{3/2}}{3\epsilon}\right), \quad y_2(x) \sim x^{-1/4}\exp\left(-\frac{2x^{3/2}}{3\epsilon}\right)$$

These represent the two dominant behaviors of the solution in the limit $\epsilon \to 0$. The exponential terms capture rapid growth or decay, while the $x^{-1/4}$ prefactor corrects the amplitude to leading order. The general solution is a linear combination of these modes that satisfies the boundary conditions.

The general solution is $y(x) \approx c_1y_1(x) + c_2y_2(x)$. We apply the initial conditions at $x = 1$. Using the WKB solutions at $x = 1$:

$y_1(1) = 1^{-1/4}\exp\left(\frac{2(1)^{3/2}}{3\epsilon}\right) = e^{2/(3\epsilon)}$, and $y_2(1) = 1^{-1/4}\exp\left(-\frac{2(1)^{3/2}}{3\epsilon}\right) = e^{-2/(3\epsilon)}$

Using the leading-order WKB derivative approximation $y'(x) \approx \frac{S_0'(x)}{\epsilon}y(x) = \frac{\sqrt{x}}{\epsilon}y(x)$:

$$y_1'(1) \approx \frac{\sqrt{1}}{\epsilon}y_1(1) = \frac{1}{\epsilon}e^{2/(3\epsilon)}$$

$$y_2'(1) \approx -\frac{\sqrt{1}}{\epsilon}y_2(1) = -\frac{1}{\epsilon}e^{-2/(3\epsilon)}$$

Applying the initial conditions $y(1) = e^{2/(3\epsilon)}$ and $y'(1) = \frac{1}{\epsilon}e^{2/(3\epsilon)}$:
$$y(1) = c_1y_1(1) + c_2y_2(1)$$
$$e^{2/(3\epsilon)} = c_1e^{2/(3\epsilon)} + c_2e^{-2/(3\epsilon)}$$

Divide the first equation by $e^{2/(3\epsilon)}$:
$$1 = c_1 + c_2e^{-4/(3\epsilon)}$$

Applying the second initial condition:
$$y'(1) = c_1y_1'(1) + c_2y_2'(1)$$
$$\frac{1}{\epsilon}e^{2/(3\epsilon)} \approx c_1\left(\frac{1}{\epsilon}e^{2/(3\epsilon)}\right) + c_2\left(-\frac{1}{\epsilon}e^{-2/(3\epsilon)}\right)$$

Divide the second equation by $\frac{1}{\epsilon}e^{2/(3\epsilon)}$:

$$1 \approx c_1 - c_2 e^{-4/(3\epsilon)}$$

For small $\epsilon$, $e^{-4/(3\epsilon)}$ is extremely small. To leading order in $\epsilon$: $1 = c_1 + c_2 \cdot$ (very small number) $\implies c_1 \approx 1$ $1 = c_1 - c_2 \cdot$ (very small number) $\implies c_1 \approx 1$ Substituting $c_1 \approx 1$ into the first equation gives $1 = 1 + c_2 e^{-4/(3\epsilon)}$, which implies $c_2 e^{-4/(3\epsilon)} = 0$. Since $e^{-4/(3\epsilon)} \neq 0$, we must have $c_2 = 0$.

Thus, these specific initial conditions select $c_1 \approx 1$ and $c_2 = 0$. The resulting solution is approximately $y(x) \sim y_1(x)$.

The specific solution satisfying these initial conditions is therefore the positive exponential branch:

$$\boxed{y(x) \sim x^{-1/4} \exp\left(\frac{2x^{3/2}}{3\epsilon}\right)}$$

### A.1.4 Asymptotic series problems

We described three types of problems in Section 3 that were the hardest for LLMs to solve. In addition to these three problem types, we included other kinds of problems that LLMs found challenging.

First, we included integrals $I(x)$ which can be approximated by an asymptotic series in the limit $x \to x_0$, for some fixed $x_0 \in \mathbb{R} \cup \{\pm\infty\}$. We find a series $\sum_{n=0}^{\infty} a_n(x - x_0)^n$ such that

$$I(x) - \sum_{n=0}^{N} a_n(x - x_0)^n << (x - x_0)^N$$

in the limit $x \to x_0$, for $N$ fixed, though we do not require the difference to converge as $N \to \infty$. These asymptotic formulas often represent expansions around essential singularities. For example, consider the integral

$$I(x) = \int_0^\infty \frac{1}{1 + x^2 t} e^{-t} dt$$

in the limit $x \to 0$. Using integration by parts to expand the integral we find that $\int_0^\infty \frac{1}{1+x^2t} e^{-t} dt \approx \sum_{n=0}^{\infty} (-1)^n n! x^{2n}$, where we can check that the right-hand side is an asymptotic series.

See an example of this technique below.

---

**Sample Asymptotic Series Problem and Full Solution**

**Problem:** Write the first two terms of the asymptotic series expansion of

$$I(x) = \int_1^x \ln(xt^2)\cos(t^3)dt$$

in the limit $x \to \infty$.

---

**Solution:** We will develop an asymptotic series using integration by parts. Define

$$u = \frac{\ln(xt^2)}{3t^2} \qquad \text{and} \qquad v = \sin(t^3).$$

Then

$$du = \frac{-2(\ln(xt^2) - 1)}{3t^3} \qquad \text{and} \qquad dv = 3t^2 \cos(t^3).$$

The formula

$$\int u dv = uv - \int v du.$$

gives us

$$I(x) = \left[\frac{\ln(xt^2)\sin(t^3)}{3t^2}\right]_1^x + \int_1^x \frac{2(\ln(xt^2) - 1)\sin(t^3)}{3t^3} dt.$$

We can apply integration by parts again to the remainder with

$$u = \frac{2(\ln(xt^2) - 1)}{9t^5} \qquad \text{and} \qquad v = -\cos(t^3)$$

and their derivatives

$$\mathrm{d}u = \frac{-2(5\ln(xt^2) - 7)}{9t^6} \qquad \text{and} \qquad \mathrm{d}v = 3t^2 \sin(t^3) \, .$$

Then we obtain

$$I(x) = \left[ \frac{\ln(xt^2)\sin(t^3)}{3t^2} - \frac{2(\ln(xt^2) - 1)\cos(t^3)}{9t^5} \right]_1^x - \int_1^x \frac{2(5\ln(xt^2) - 7)\cos(t^3)}{9t^6} \mathrm{d}t \, .$$

Therefore, the first two terms of the asymptotic series expansion are

$$\boxed{\frac{\ln(x^3)\sin(x^3)}{3x^2} - \frac{\ln(x)\sin(1)}{3} - \frac{2(\ln(x^3) - 1)\cos(x^3)}{9x^5} + \frac{2(\ln(x) - 1)\cos(1)}{9}}.$$

### A.1.5 Integral problems

In addition to asymptotic series problems, we include a broader class of one-dimensional parametric integrals of the form $I(\lambda) = \int_a^b \phi(\lambda; x)dt$, where the integrand $\phi(\lambda; x)$ may involve elementary functions, special functions, parameter-dependent exponents, singularities, polynomial or rational prefactors, or oscillations. These integrals are parameterized by a variable $\lambda$ that controls the problem's asymptotic behavior.

We are particularly interested in Laplace-type integrals with the form

$$I(\lambda) = \int_a^b f(x)e^{-\lambda g(x)}dx$$

We wish to construct analytical estimates $\phi_n(\lambda)$ of $I(\lambda)$ such that for each $n$, $\phi_n(\lambda)$ behaves similar to $I(\lambda)$ as $\lambda \to \infty$ for each $n$, and then find the optimal $N$ with estimate $\phi_N$.

We first provide an example of an integral that can be solved using an ordinary taylor approximation.

---

**Sample Integral Problem and Full Solution**

**Problem:**
Find the leading behavior up to $O(x^6)$ as $x \to 0_+$ of

$$I(x) = \int_x^1 \cos(xt)dt.$$

(Problem taken from Bender and Orszag [2013].)

---

**Solution:**
In finding the leading behavior of an integral
$I(x) = \int_a^b f(t, x)dt$ as $x \to x_0$
If $f(t, x) \sim f_0(t), x \to x_0$ uniformly in the interval $a \leq t \leq b$, then
$I(x) = \int_a^b f(t, x)dt \sim \int_a^b f_0(t)dt$ as $x \to x_0$
The function $\cos(xt)$ can be approximated for small $x$ with a Taylor series expansion

$$\cos(xt) = 1 - \frac{(xt)^2}{2!} + \frac{(xt)^4}{4!} - \frac{(xt)^6}{6!} \cdots$$

This series converges uniformly for $0 \leq x \leq t \leq 1$, so we can integrate the terms in the Taylor series expansion in order to determine the leading behavior of this integral.

$$\int_x^1 \cos(xt)dt \sim \int_x^1 \left(1 - \frac{(xt)^2}{2!} + \frac{(xt)^4}{4!} - \frac{(xt)^6}{6!}...\right)$$

$$= (1-x) - \frac{1}{2}x^2 \left(\frac{1}{3} - \frac{x^3}{3}\right) + \frac{1}{24}x^4 \left(\frac{1}{5} - \frac{x^5}{5}\right) - \frac{1}{720}x^6 \left(\frac{1}{7} - \frac{x^7}{7}\right)...$$

The leading order behavior as $x \to 0+$ for the integral is given by

$$I(x) = (1-x) - \frac{1}{2}x^2 \left(\frac{1}{3} - \frac{x^3}{3}\right) + \frac{1}{24}x^4 \left(\frac{1}{5} - \frac{x^5}{5}\right) - \frac{1}{720}x^6 \left(\frac{1}{7} - \frac{x^7}{7}\right)$$

The solution up to order 6

$$\boxed{I(x) = 1 - x - \frac{x^2}{6} + \frac{x^4}{120} + \frac{x^5}{6} - \frac{x^6}{5040}}$$

We now provide an example applying Laplace's method to solve an integral.

---

## Sample Integral Ratio Problem and Asymptotic Solution

**Problem:**
Estimate the leading-order behavior as $x \to \infty$ of the ratio

$$\left| \frac{\int_0^\infty \frac{t^{x-1}}{t+x} e^{-t^{1/4}} dt}{\int_0^\infty t^{x-1} e^{-t^{1/4}} dt} \right|$$

---

**Solution:**
To understand the behavior of the integrals, plot the function $t^{x-1}e^{-t^{1/4}}$ for large $x$; it becomes sharply peaked around a point $t^*$. This localization allows us to approximate the slowly varying factor $\frac{1}{t+x} \approx \frac{1}{t^*+x}$, and pull it out of the numerator integral:

$$\left| \frac{\int_0^\infty \frac{t^{x-1}}{t+x} e^{-t^{1/4}} dt}{\int_0^\infty t^{x-1} e^{-t^{1/4}} dt} \right| \approx \left| \frac{1}{t^*+x} \cdot \frac{\int_0^\infty t^{x-1} e^{-t^{1/4}} dt}{\int_0^\infty t^{x-1} e^{-t^{1/4}} dt} \right| = \left| \frac{1}{t^*+x} \right|$$

To locate the peak, define:

$$\phi(t) = \ln(t^{x-1}) - t^{1/4} = (x-1)\ln t - t^{1/4}$$

and solve $\phi'(t^*) = 0$ for the maximum:

$$\phi'(t) = \frac{x-1}{t} - \frac{1}{4}t^{-3/4} \quad \Rightarrow \quad \frac{x-1}{t^*} = \frac{1}{4}(t^*)^{-3/4}$$

Multiplying both sides by $t^*$:

$$x - 1 = \frac{1}{4}(t^*)^{1/4} \quad \Rightarrow \quad t^* = (4(x-1))^4 \approx (4x)^4$$

Thus:

$$\left| \frac{1}{t^*+x} \right| \approx \frac{1}{(4x)^4} = \frac{1}{256\,x^4}$$

Final Answer:

$$\boxed{\left| \frac{\int_0^\infty \frac{t^{x-1}}{t+x} e^{-t^{1/4}} dt}{\int_0^\infty t^{x-1} e^{-t^{1/4}} dt} \right| \sim \frac{1}{256\,x^4} \quad \text{as } x \to \infty}$$

### A.2 Evaluation setup

#### A.2.1 Preparation of problem and solution

Each problem in the dataset went through a thorough verification procedure. For each problem generated by one student, another student was instructed to work through the same problem and ensure their answer matched the original solution. Figure 5 provides a visual comparison of the numerical and approximate solutions for the boundary value problem in Appendix A.1.2. This allows for semi-automated human verification that analytical solutions correspond well with numerical ground-truths, a method used to verify the problems in the dataset. Finally, students made sure that all of the problem statements and solutions followed consistent formatting guidelines that could be easily parsed and compared to the model responses.

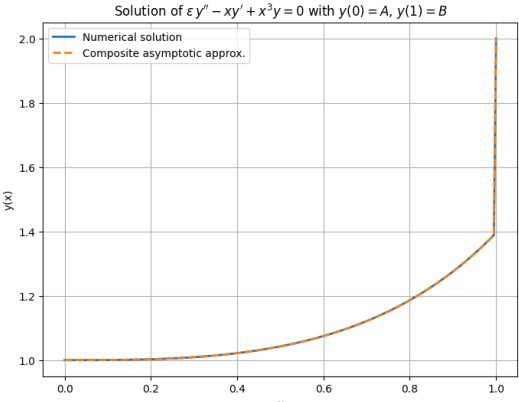

Figure 5: Visual comparison of numerical and approximate analytical solutions to a sample boundary value problem for solution verification.

#### A.2.2 Prompts for response generation

To ensure that model responses to the questions weren't driven by particular question wordings, we aimed to standardize prompts as much as possible based on the question type. Note that for the nonlinear PDE questions, there was some variance in the question types and so a representative task instruction (for when a problem asked for a self-similarity solution) is included below. If the nonlinear PDE questions additionally involved initial or boundary conditions, those were incorporated into the problem statement.

The prompts were combined with the following prompt suffix: Place your final answer in a `\boxed{}` LaTeX environment. If you have multiple answers, separate them with a ";". Use the notation from the problem and do not define any new variables.

#### A.2.3 Numerical evaluation

Part of the importance of having a standardized prompt and response format was to ensure consistency in numerical evaluation of the solutions provided by students and the various models. Provided `\boxed{}` LaTeX solutions were parsed into symbolic representations in a method discussed in A.4. These symbolic representations were then evaluated at a particular value for all of the input variables into the problem. For example, if a problem had its solution in terms of the variable $x$, the solution was evaluated at a particular value for $x$. The model solutions were then graded whether at that value they numerically matched the verified student solutions.

### A.3 Rubrics for LLM-as-a-judge evaluation

While numerical or symbolic evaluations are now the gold standard in mathematical benchmarks, they do not capture the problem-solving process of models. Oftentimes, a model can get close to the ground-truth solution but is slightly off—it can be useful for the pass rates to reflect this information,

Table 4: Prompts by Question Type

| Question Type | Inputs | Task instruction |
|---|---|---|
| WKB Approximation | Problem, Initial Conditions | Find the leading order WKB approximation for the specific differential equation `{Problem}` with initial conditions at `{Initial Conditions}` where $\epsilon$ is a small positive parameter ($0 < \epsilon \ll 1$). Use only the variables and constants given in the problem; do not define additional constants. Place your final solution in a \boxed{} LaTeX environment. |
| Integral | Problem, Limit | Consider the following integral: `{Problem}` In the limit `{Limit}`, find approximate behavior of the integral up to leading non-zero order in $\epsilon$. Provide your answer in a {\boxed{}} LaTeX environment. |
| Nonlinear ODE | Problem, Limit | Find the leading order behavior of `{Problem}` in the limit `{Limit}`. Please place your final solution in a {\boxed{}} LaTeX environment. |
| Boundary Layer | Problem, Boundary Conditions | Find a uniformly valid approximation to the solution of `{Problem}` with boundary conditions `{Boundary Conditions}` in the limit $\epsilon \ll 1$. Use only the variables and constants given in the problem; do not define additional constants. Place your final solution in a \boxed{} LaTeX environment. |
| Nonlinear PDE | Problem, Limit | Find a self similarity solution for the non-linear partial differential equation `{Problem}` in the limit `{Limit}`. Please place your final solution in a \boxed{} LaTeX Environment. |
| Asymptotic Series | Problem, Limit | Find the first two terms in the asymptotic series of `{Problem}` in the limit `{Limit}`. Provide your answer in a \boxed{} LaTeX environment. |

especially in a pedagogical setting. Below, we present the rubrics used to produce LLM-as-a-judge accuracy results for WKB problems, boundary layer problems, and nonlinear PDEs.

---

**Grading rubric for boundary layer problems (5 points)**

1. Recognition of Boundary Layer Structure
- 1 point: Explicitly states that boundary layer analysis is needed due to a small parameter multiplying the highest derivative or a rapid solution change near a boundary, and names the parameter.
- 0 points: Does not state that boundary layer analysis is required, does not identify a small parameter, or gives an otherwise incorrect reason.

2. Identification of Boundary Layer Location
- 1 point: Explicitly identifies the correct boundary (e.g., $x = 0$ or $x = 1$) where the boundary layer occurs, and justifies this location.

---

- 0 points: Does not identify a boundary layer location, selects the wrong location, or does not justify the choice.

3. Scaling and Inner Variable (1 point)
- 1 point: Writes down the correct inner variable (e.g., $\xi = x/\epsilon$), substitutes it into the equation, and obtains the correctly rescaled inner equation.
- 0 points: Student does not define the correct inner variable, does not substitute correctly, or otherwise does not derive the correct inner equation.

4. Outer and Inner Solutions (1 point)
- 1 point: Writes the correct general outer solution (with small parameter set to zero) and the correct general inner solution (to the rescaled equation), including arbitrary constants.
- 0 points: Omits the general form of either solution, does not include arbitrary constants, or writes an otherwise incorrect solution.

5. Matching and Composite Solution (1 point)
- 1 point: Matches inner and outer solutions in the overlap region, determines all constants, and writes the correct uniformly valid composite solution (e.g., $y_{\text{comp}}(x) = y_{\text{outer}}(x) + y_{\text{inner}}(x) - y_{\text{overlap}}(x)$).
- 0 points: Does not expand and match solutions, does not determine constants, or otherwise does not write the correct composite solution.

Note: you should only award a 5/5 if the model's final solution EXACTLY matches the ground-truth solution.

## Grading rubric for WKB problems (5 points)

1. Recognition of WKB applicability
- 1 point: Clearly recognizes that the equation is suitable for WKB (i.e., contains a small parameter multiplying the highest derivative or rapidly varying solution), and explains why WKB is appropriate (e.g., discusses scales, nature of coefficients, physical context).
- 0 points: Fails to recognize the need for WKB, uses an inappropriate method, or provides weak justification for why WKB is needed.

2. Correct Statement of the WKB ansatz
- 1 point: States the correct WKB ansatz (e.g., $y(x) \sim \exp\left[\frac{1}{\epsilon}S(x)\right]$ or similar), and includes all relevant assumptions (e.g., expansion of $S(x)$, small parameter $\epsilon$).
- 0 points: Ansatz is missing or incorrect.

3. Substitution and Derivation of the WKB Equation
- 1 point: Correctly substitutes the ansatz into the original equation, carefully computes all derivatives, and systematically derives the WKB equation with all steps shown.
- 0 points: No meaningful substitution or substitution/derivation is incorrect.

4. Dominant Balance and Leading Order Equation
- 1 point: Correctly identifies the dominant balance in the WKB equation (e.g., recognizes that the $(S')^2$ term dominates for small $\epsilon$), writes the leading-order equation, and justifies neglecting smaller terms.
- 0 points: Fails to identify dominant balance or applies an otherwise incorrect dominant balance.

5. Solution for Leading Order and Interpretation
- 1 points: Correctly solves the leading order equation for $S(x)$, writes the leading order solution for $y(x)$, and interprets/expresses the result in the correct form, including proper handling of constants or boundary conditions if needed.
- 0 points: Solution is missing, misinterpreted, or otherwise incorrect.

Note: you should only award a 5/5 if the model's final solution EXACTLY matches the ground-truth solution.

> **Grading rubric for nonlinear PDE problems (5 points)**
>
> 1. Recognition of the Role of Nonlinearity
> - 1 point: Explicitly recognizes that the equation is nonlinear and states why the nonlinear term is essential for the solution or asymptotic behavior.
> - 0 points: Does not recognize nonlinearity, does not discuss the nonlinear term, or gives an otherwise incorrect justification.
>
> 2. Ansatz or Similarity Variable
> - 1 point: States the correct ansatz, similarity variable, or reduction (e.g., traveling wave substitution $u(x,t) = f(x - ct)$, similarity variable for blow-up, or other appropriate reduction for the nonlinear PDE).
> - 0 points: Does not write a correct ansatz, similarity variable, or reduction, or otherwise makes an incorrect substitution.
>
> 3. Substitution and Reduction to ODE
> - 1 point: Correctly substitutes the ansatz or similarity variable into the original PDE and reduces it to the appropriate ODE, including all necessary algebraic steps.
> - 0 points: Does not substitute correctly, does not reduce to the correct ODE, or otherwise makes an error in the reduction.
>
> 4. Solution of the Reduced ODE
> - 1 point: Correctly solves the reduced ODE, including all arbitrary constants or relevant parameters, and writes the general or particular solution as required.
> - 0 points: Solution is missing, incorrect, omits constants/parameters, or otherwise does not solve the reduced ODE correctly.
>
> 5. Interpretation and Asymptotic/Physical Behavior
> - 1 point: Correctly interprets the solution in terms of the original variables (e.g., describes soliton or blow-up, gives the asymptotic or qualitative behavior), and addresses any conditions or parameter regimes relevant to the original nonlinear PDE.
> - 0 points: Fails to interpret, gives an incorrect physical/asymptotic description, or otherwise does not address the original nonlinear PDE's physical meaning.
>
> Note: you should only award a 5/5 if the model's final solution EXACTLY matches the ground-truth solution.

## A.4 Automated parsing and model evaluation

To compare LLM-generated solutions against ground-truth solutions written by students, our parser converts LaTeX expressions into symbolic representations that can be programmatically evaluated for numerical closeness.

The parser architecture is designed to handle complexities and variations in mathematical notation. Initially, the system extracts solutions from LaTeX \boxed{} environments using regular expression pattern matching. It also is able to process multiple solutions using the semicolon as a delimiter. This extraction ensures that only the final answers are evaluated, filtering out intermediate text. After the extraction, the system converts LaTeX notation into SymPy expressions through a series of transformation rules, such as replacing Unicode characters with LaTeX code and removing unnecessary formatting. The parser then processes specialized symbols like integrals, beta functions, and expressions with superscripts and subscripts to translate them into a standardized format.

All models are prompted to provide their final answer in a LaTeX \boxed{} environment. Then, a custom-built parser uses Python's RegEx library to search for the final solution at the end of the model's output. The parser then converts the extracted LaTeX expression into a SymPy expression.

In our types of applied math problems, it is possible for different functional forms to simultaneously be valid approximations; therefore, we check correctness by evaluating the model's SymPy expression and the ground-truth solution produced by students—which is also converted into a SymPy expression—at the same point in the domain and determining whether the values are within a closeness threshold. This approach to evaluation ensures that the model's solution is considered correct only if it precisely matches the ground-truth solution. We ensure that there is no ambiguity in the prompts to the model by specifying any variables or free parameters—therefore, all problems can be answered as a function only of their independent variables.

### A.5  Additional analysis of model failure modes

Given that our evaluation framework strictly required solutions in a `\boxed{}` environment, we noticed that some models failed to follow these instructions. In particular, DeepSeek-R1 never placed its final solution in a LaTeX box. Our first instinct was that the model was exceeding its max tokens, which is set by default to 32,768 tokens. However, we found that the model was rarely reaching the limit and simply failed to converge to a final answer written in proper mathematical formatting. The model also often replaced mathematical expressions with LaTeX ellipses (specifically, $\ldots$), presumably for convenience, but never plugged the true mathematical symbols or expressions back in to its final result. Therefore, we did not include the model in our evaluation results.

Other models also exhibited difficulty following the instructions in our prompts. In particular, despite its high performance on our benchmark, Gemini 2.5 Pro frequently ignored formatting guidelines, using different LaTeX formats to flag its final solution. This is one potential explanation for why Gemini 2.5 Flash Thinking showed higher overall accuracy than Gemini 2.5 Pro, despite the latter supposedly being Google's most performant model.

