# OpenReview forum: "HARDMath2: A Benchmark for Applied Mathematics Built by Students as Part of a Graduate Class"
_NeurIPS.cc/2025/Datasets_and_Benchmarks_Track — NeurIPS 2025 Datasets and Benchmarks Track poster_

### Official Review · Reviewer_SwXX · 2025-06-22

**Rating:** 5
**Confidence:** 3

**Summary:**

The paper proposed a dataset of 211 original problems covering core topics in applied mathematics. It fills the gap of the lack of benchmark dataset for LLM in the field of computational science and engineering. The dataset is constructed in a collaborative environment where the level of difficulties are refined.

**Dataset Code Accessibility:**

Yes

**Ethical Considerations:**

No, there are no or only very minor ethics concerns

**Final Justification:**

I appreciate the authors' hard work at responding, including the extra results using the detailed rubrics from grading the students in the course to provide partial-credit scoring using an LLM-judge. This updated evaluation metrics make a lot more sense.
After explanation, I understand that given the current existing size of the datasets in this field and the importance of asymptotic methods in applied math/physics, the current dataset size is acceptable. However, the authors should consider expanding the dataset and automated creation method in the future.
Hence, I have decided to increase my score.

**Limitations Weaknesses:**

Authors may consider enlarging the dataset further to include a wider range of topics and categories of problems from applied math. Although the dataset comprises 211 original problems, it only covers a small subset of applied math problems.

While the dataset creation process appears to be an interesting approach, it also seems very time- and energy-consuming. The paper mentions common failure modes identified through interactions with the system. It might be worth exploring whether the full dataset can be constructed more efficiently, for example, by introducing certain obfuscation techniques.

I think the interactive setting could be a valuable tool for identifying different failure modes across various large models. In the paper, the authors only include one model for interaction, which may limit the results that the common failure modes observed could be specific to that particular model.

**Strengths Contributions:**

The paper focuses on challenging questions from applied mathematics involving a variety of approximation-based problems.
The creation process of the dataset also seems interesting as it is a useful strategy to create increasingly difficult problems targeting on the failure modes of LLM through interaction with the models.
Given the interactive nature of the dataset creation process, all problems are original and safe from data contamination.

---

> ### Author Rebuttal · Authors · 2025-07-31
>
> Thank you for your thoughtful review! Your comments on our paper’s strengths and areas to improve are appreciated. Here are our responses to your feedback, as well as some new results to incorporate your suggestions.
>
> **1. ```Authors may consider enlarging the dataset further to include a wider range of topics and categories of problems from applied math```**
>
> We believe that our benchmark actually is quite large and diverse compared to similar mathematical and quantitative benchmarks. It is true that asymptotic methods are used in most of the problems in our benchmark, but we would like to emphasize that they are absolutely central to applied mathematics as a field. Since closed-form solutions do not exist for the majority of problems in applied mathematics, the field by definition requires approximations that only hold in certain regimes. Therefore, we believe the problems covered in this benchmark—spanning various forms of PDEs, ODEs, stochastic differential equations, complex integrals, and series expansions—comprise the bulk of applied mathematics relevant to the physical sciences and engineering. In particular, this benchmark was developed in a university course modeled after the canonical textbook *Advanced mathematical methods for scientists and engineers* by Bender & Orzsag [1] used as the foundation of many graduate courses, as well as the field of physical mathematics.
>
> We also *intentionally excluded* several classes of problems covered in the class that we considered “too easy.” We tested frontier models on other problems like dimensional analysis, simple integrals, and higher-order polynomials, and found that frontier LLMs have mostly saturated such problems. Therefore, we believe that the benchmark in fact covers a diverse range of techniques that are challenging.
>
> Regarding the size of the dataset, we would like to note that the only benchmarks comparable to HARDMath2 in terms of difficulty and sophistication are generally much smaller. For example, one such benchmark is the Theoretical Physics Benchmark (TPBench) [2], which includes only 36 problems at the “easy” graduate level or above. Similarly, MathArena [3], which is now commonly used to evaluate frontier LLMs only includes problems directly taken from public mathematical olympiads. As the IMO and USAMO both included only six problems each, we in fact believe our benchmark is quite large. Finally, FrontierMath [4] includes 350 problems, which is only slightly larger than HARDMath2. However, the creators of FrontierMath note that their problems were written by math professors and postdocs, each hired for a several-week-long research project resulting in one new problem for their benchmark, with funding from OpenAI. In contrast, **HARDMath2 was created entirely within the homework assignments of a university course, requiring no monetary support or professor-level problem-writers.**
>
> We believe our framework is a significant contribution and can be easily applied in other graduate-level courses to build high-quality benchmarks while simultaneously improving learning outcomes for students.
>
> **2. ```Exploring whether the full dataset can be constructed more efficiently```**
>
> In contrast to the first HARDMath benchmark, which was created programmatically, HARDMath2 was manually curated to increase problem diversity and difficulty. Currently, there do not seem to be any methods to scale dataset creation for the level of sophistication involved in our benchmark. Other comparable benchmarks like TP Bench, FrontierMath, and Humanity’s Last Exam [6] all require manual curation (with the latter two compensating problem-writers), while older datasets simply involve scraping problems from existing sources.
>
> Further, much of the instructor-level intervention that was necessary was a result of the novelty of the method and the lack of available infrastructure at the beginning of the course. By the last assignment, when the infrastructure was more mature and the students familiar with the process, generation of over 100 potential PDE problems (not all of which were used) required less than one hour of instructor intervention. Given the success we have seen in improving pedagogical outcomes and the relative ease of applying these methods with a more mature system, we plan to have students in the next iteration of the course this fall continue to expand on the benchmark.
>
> That being said, there is still substantial room to improve the scalability of our dataset and are providing our framework to systematically make the problems more difficult, which we describe at the bottom of this message.
>
> We hope that this provides more insight into the method by which we made problems more challenging for the models. If the paper is to be accepted, we will certainly add a new section detailing this process.
>
> **3. ```The authors only include one model for interaction, which may limit the results that the common failure modes observed could be specific to that particular model.```**
>
> Our Google Sheets framework actually used four models: Gemini 2.0 Flash, Gemini 2.0 Flash Thinking, Gemini 2.5 Pro, and OpenAI o1. We used 2.0 Flash as the minimum threshold; if a problem submitted by a student was solvable by this model, the student would need to modify it to make it more difficult. However, the other three models were all automatically run on those problems as well to allow students to gauge their problem’s difficulty and understand the models' different failure modes. We could add any model to the spreadsheet but chose this selection to minimize API costs, since students might be querying the API many times per problem while developing the problems.
>
> ______
>
> **Framework for systematically increasing the difficulty of problems**
>
>
>
> We begin with an "easy" problem that a weaker model like Gemini 2.0 can already solve as its initial input. Our framework relies on two instances of a powerful model (Gemini 2.5 Pro): one instance acts as both an all-knowing verifier and obfuscator, which has access to the original problem, the original solution, and all modifications to the original problem statement, while the second instance acts as a solver to test the difficulty of the new problem statement. The verifier first modifies the problem and passes it to the (context-less) solver, which attempts to solve the modified problem. The verifier, which knows the modification that was made, will check the solution generated by the solver and assess its correctness. If it deems the solution to be correct, it will again modify the problem statement, but if it finds the solution incorrect, we end the process there, generate a solution from the verifier, and manually check the solution provided by the model. Students found that the verifier was typically quite skilled at identifying flaws in the solver’s proposed solution.
>
> We use two classes of techniques that either leave the original solution unchanged or modify the solution. Methods to retain the original solution include the addition of cancellable terms, replacement of simple terms with other terms that are equivalent but algebraically more complex, and multiplication by a complex function that evaluates to 1 in the limit of interest. These do not require the verifier model above, as we can be certain that the original solution does not change and just check whether the solver’s solution matches the original solution. Alternatively, we can change the solution of the problem itself by adding or composing trigonometric, logarithmic, hyperbolic, or other special functions in the problem (e.g., $f(x)$ becomes $sin(f(x))$. These problems are typically more difficult for the models to solve, and we require the human-in-the-loop to resolve the new problem to guarantee the solution’s correctness.
> We hope these additional details and results address your concerns. Thank you again for your attention!
> ____________
>
> References
>
> [1] Chung et al. (2025), Theoretical Physics Benchmark (TPBench) - a Dataset and Study
> of AI Reasoning Capabilities in Theoretical Physics
>
> [2] Balunović et al. (2025), MathArena: Evaluating LLMs on Uncontaminated Math Competitions
>
> [3] Bender and Orszag (1978). Advanced Mathematical Methods for Scientists and Engineers
>
> [4] Glazer et al. (2025), FrontierMath: A Benchmark for Evaluating Advanced Mathematical Reasoning in AI

---

> ### Comment · Area_Chair_75KT · 2025-08-03
>
> Dear reviewer,
>
> Please read the rebuttal and provide your *final justification* and score.
>
> Best,
>
> AC

---

> ### Comment · Reviewer_SwXX · 2025-08-04
>
> I appreciate the authors' hard work at responding, including the extra results. I also have a better understanding of the current state of benchmark dataset in this specific field. I have therefore raised my score by 1.

---

> > ### Author Response · Authors · 2025-08-06
> >
> > Thank you very much! We appreciate your feedback.

---

### Official Review · Reviewer_9dGN · 2025-06-30

**Rating:** 4
**Confidence:** 4

**Summary:**

HARDMATH2 introduces a benchmark with 211 original graduate-level applied math problems, covering mainly asymptotic methods and PDE/ODE analysis. It is curated via a live course where students iteratively propose, refine, and verify their own questions in response to LLM outputs. Each question is specified in LaTex and can be parsed and validated numerically.

**Additional Feedback:**

I recommend weak accept for the paper. Although the benchmark has limitations in scale and topic coverage—focusing mainly on introductory applied mathematics covering PDEs and ODEs—the integration of graduate-level coursework with dataset curation shows an interesting contribution. It can be possibly extended to  to additional graduate courses across other applied mathematics domains to provide more comprehensive evaluation of LLMs. Should the paper be accepted, I encourage the authors to expand the dataset in future semesters to include broader mathematical topics and increase the benchmark's scope.

**Dataset Code Accessibility:**

Yes

**Dataset Code Comments:**

Data is available on huggingface and code for evaluation is also submitted to Github.

**Ethical Considerations:**

No, there are no or only very minor ethics concerns

**Final Justification:**

As I mentioned in my response, there are already many math benchmarks. Hence, I tend to keep my original rating.

**Limitations Weaknesses:**

1. Modest scale. The scale is limited with only 211 problems. It may be underpowered compared to larger benchmarks.
2. Limited domain coverage. Focuses exclusively on asymptotic PDE/ODE methods; other essential graduate-level topics are not included.
3. Human effort requirements. The current pipeline relies heavily on instructor and TA oversight, which could impede broader adoption.

**Strengths Contributions:**

1. High-quality and domain-specific. The problems are targeting critical techniques in applied math and scientific computing but less-studied in existing LLM benchmarks.
2. Innovative course-integrated curation. Although the dataset itself is not very innovative, the integration with a live graduate course is interesting.
3. The evaluation pipeline is reproducible and automated. The LaTex, SymPy, and numerical check pipeline avoids human grading, making evaluation robust and reproducible.

---

> ### Author Rebuttal · Authors · 2025-07-31
>
> Thank you very much for your feedback on our paper! We are glad that you find our benchmark high-quality and the curation method innovative. Here are our responses and new results to incorporate your suggestions.
>
> **1. ```Modest scale```**
>
> We would like to note that the only benchmarks that are closely comparable to HARDMath2 in terms of difficulty and sophistication are generally much smaller. For example, one such benchmark is the Theoretical Physics Benchmark (TPBench) [1], which includes only 36 problems at the “easy” graduate level or above. Similarly, MathArena [2], which is now commonly used to evaluate frontier LLMs only includes problems directly taken from public mathematical olympiads. As the IMO and USAMO both included only six problems each, we in fact believe our benchmark is quite large relative to similar benchmarks in this space.
>
> **2. ```Limited domain coverage```**
>
> While asymptotic methods are indeed at the heart of most problems covered in our benchmark, we would like to emphasize that they are absolutely central to applied mathematics as a field. Since closed-form solutions do not exist for the majority of problems in applied mathematics, the field by definition requires approximations that only hold in certain regimes. (Also, it is worth noting that the dataset was developed in a university course modeled after the canonical textbook *Advanced mathematical methods for scientists and engineers* by Bender & Orzsag [3].)
>
> The dataset also includes many problems that are not PDEs or ODEs, such as asymptotic series, stochastic differential equations, and various types of complex integrals. But importantly, we’d like to point out again that these problem types cover the bulk of the fundamentals of applied mathematical techniques relevant to the physical sciences and engineering, and indeed represent the entire curriculum of the core applied mathematics course for graduate students at our university. We also *intentionally excluded* several problem types covered in the class from the benchmark because we found that frontier models have saturated such problems (e.g., dimensional analysis, simple integrals, and high-order polynomials) and did not see a point in including problems that the best LLMs can already consistently solve. Therefore, we believe that the benchmark in fact covers a diverse range of problems and techniques.
>
> Furthermore, one of the aims of our benchmark is to provide a template for others to use in constructing similar benchmarks covering a wider range of advanced, graduate-level topics. We believe that the pedagogical approach we took to developing this benchmark enables both better learning outcomes in students and could easily be implemented in other graduate-level courses to develop similar, high-quality fundamental benchmarks with a fraction of the time and expense incurred by other leading benchmarks in this area.
>
> **3. ```Human effort requirements```**
>
> The goal of releasing this benchmark was twofold: to provide a new mathematical benchmark that current LLMs find challenging and to introduce a novel approach for dataset curation. It is true that our benchmark was manually developed, but unlike other benchmarks that require hiring experts to write and solve new problems or scraping already existing resources, we develop original problems via a pedagogical approach.
>
> Broadly speaking, our perspective is that by inventing new problems (*which current AI cannot solve*), students will learn significantly more than the traditional approach of solving homework problems prescribed by the instructor. We hope to provide a general framework that university courses can adopt to both improve the learning process of their students and develop new challenging benchmarks. Importantly, our automated framework of interactively writing problems and iteratively increasing their difficulty can be applied to any quantitative domain.
>
> While this benchmark did require instructor/TA oversight given the novelty of the approach, the system was largely automatic by the end of the course. For their final assignment, students were asked to generate and submit nonlinear PDE problems. At this point, our infrastructure was tested and students were used to the process, which resulted in minimal (less than one hour) of instructor intervention to generate over 100 potential PDE problems. At this point, we plan to continue expanding the HardMATH dataset in future semesters during future iterations of the class both to further enhance the benchmark but importantly because of the pedagogical value of this exercise for the students.
>
> Finally, we agree that it would be beneficial to continue expanding the dataset. In fact, this benchmark is already an expansion of the first HARDMath benchmark released in the previous year’s iteration of this course, in which we focused on writing harder and more diverse problems! We hope that the acceptance of this paper will encourage other course instructors to adopt our framework to improve the quality and depth of learning in their classrooms, as well as the availability of high-quality, open-source datasets for LLMs.
>
> __________
>
> References
>
> [1] Chung et al. (2025), Theoretical Physics Benchmark (TPBench) - a Dataset and Study
> of AI Reasoning Capabilities in Theoretical Physics
>
> [2] Balunović et al. (2025), MathArena: Evaluating LLMs on Uncontaminated Math Competitions
>
> [3] Bender and Orszag (1978). Advanced Mathematical Methods for Scientists and Engineers

---

> > ### Author Response · Authors · 2025-08-03
> > **Framework for systematically increasing the difficulty of problems**
> >
> > Dear reviewer,
> >
> > We also wanted to provide some additional detail on the automatic component of our problem generation framework in response to your concerns about the human effort required to create the dataset. This should make our approach even more accessible.
> >
> > __________
> >
> > When we have an "easy" problem that a weaker model like Gemini 2.0 can already solve, we can use the following framework to systematically increasing the difficulty of that problem. Our framework relies on two instances of a powerful model (Gemini 2.5 Pro): one instance acts as both an all-knowing verifier and obfuscator, which has access to the original problem, the original solution, and all modifications to the original problem statement, while the second instance acts as a solver to test the difficulty of the new problem statement. The verifier first modifies the problem and passes it to the (context-less) solver, which attempts to solve the modified problem. The verifier, which knows the modification that was made, will check the solution generated by the solver and assess its correctness. If it deems the solution to be correct, it will again modify the problem statement, but if it finds the solution incorrect, we end the process there, generate a solution from the verifier, and manually check the solution provided by the model. Students found that the verifier was typically quite skilled at identifying flaws in the solver’s proposed solution.
> >
> > We use two classes of techniques that either leave the original solution unchanged or modify the solution. Methods to retain the original solution include the addition of cancellable terms, replacement of simple terms with other terms that are equivalent but algebraically more complex, and multiplication by a complex function that evaluates to 1 in the limit of interest. These do not require the verifier model above, as we can be certain that the original solution does not change and just check whether the solver’s solution matches the original solution. Alternatively, we can change the solution of the problem itself by adding or composing trigonometric, logarithmic, hyperbolic, or other special functions in the problem (e.g., $f(x)$ becomes $\sin(f(x))$. These problems are typically more difficult for the models to solve, and we require the human-in-the-loop to resolve the new problem to guarantee the solution’s correctness.
> >
> > We hope this addresses your concerns! We plan to detail this more in our final manuscript should the paper be accepted.

---

> > ### Comment · Reviewer_9dGN · 2025-08-08
> >
> > Thank you for your detailed rebuttal. I agree that developing original problems via a pedagogical approach is a valuable direction for strengthening LLMs in the future. However, given the existing abundance of (hard) math datasets, I decided to keep my original rating.

---

> > > ### Author Response · Authors · 2025-08-08
> > > **Uniqueness of applied mathematics datasets**
> > >
> > > Thank you for your response!
> > >
> > > We do want to re-emphasize that there are no other datasets that cover this broad category of mathematics at this difficulty or size. Existing benchmarks like FrontierMath, the IMO, and the USA Mathematical Olympiad (USAMO) only cover topics in "formal/pure" mathematics or competition-style problems. They do not cover **applied mathematics**, which is absolutely critical for science and engineering. In particular, none of our problems admit closed-form exact solutions, which necessitates deep knowledge of mathematical approximation techniques, while other benchmarks are almost all exactly solvable.
> > >
> > > Finally, as you mentioned in your original review, our problems target ```critical techniques in applied math and scientific computing but less-studied in existing LLM benchmarks.``` While many of the topics covered in those benchmarks are primarily useful only for mathematics research, the techniques and methods tested in **HARDMath2** are also broadly valuable outside of math. In fact, many of the students enrolled in the course were from neighboring fields such as physics, biology, mechanical engineering, and statistics.

---

> ### Comment · Area_Chair_75KT · 2025-08-03
>
> Dear reviewer,
>
> Please read the rebuttal and provide your *final justification* and score.
>
> Best,
>
> AC

---

> ### Author Response · Authors · 2025-08-06
>
> Dear reviewer, we just wanted to follow up on your feedback and ask if there's anything else we can clarify outside our replies and new results. To summarize, we believe our dataset is actually quite large and diverse compared to similar benchmarks, and covers essential material in graduate-level applied mathematics.
>
> We have also provided a description of the framework used to systematically increase the difficulty of our problems, which helps decrease the need for human problem-writers. By the end of the course, problem generation had become quite methodical, and we believe new benchmarks can easily be created in many other courses.

---

### Official Review · Reviewer_NDnL · 2025-07-01

**Rating:** 5
**Confidence:** 3

**Summary:**

This paper introduces HARDMath2, a novel benchmark designed to evaluate LLMs on applied math problems in graduate-level courses. Unlike previous benchmarks that focus on problems with exact solutions, HARDMath2 emphasizes on problems with approximation-based reasoning, including boundary-layer analysis, WKB methods, and non-linear PDEs, etc. A key innovation is its student-driven process. Students enrolled in a graduate applied math course collaboratively designed, solved, and validated problems while using LLMs both as aids and adversaries. The paper presents a detailed methodology for problem curation, validation, and automatic evaluation. Experiments show that current SOTA LLMs still struggle with tasks requiring multi-step mathematical reasoning.

**Dataset Code Accessibility:**

Yes

**Ethical Considerations:**

No, there are no or only very minor ethics concerns

**Final Justification:**

I appreciate the authors’ responses. I think this work presents a meaningful contribution, offering a valuable benchmark for evaluating the mathematical abilities of AI systems.
While I had minor reservations initially, these were resolved during the discussion phase. I maintain my positive score (5).

**Limitations Weaknesses:**

- The current evaluation explicitly compares only the final numerical answer, filtering out intermediate steps or reasoning chains. This is appropriate for objective assessment, but it disregards models that may have strong reasoning but weak final formatting, or models that partially solve a problem in a way informative to humans. The framework could potentially be augmented with partial credit metrics or human-in-the-loop validation modes.

- The paper briefly notes that some models (e.g., DeepSeek-R1) failed to follow instructions, leading to their exclusion from the evaluation. However, this raises broader concerns about instruction-following robustness and sensitivity to the designed prompt. The decision to exclude such models is understandable for numerical fairness, but a quantitative report on how many outputs were rejected due to formatting issues would provide more transparency.

**Strengths Contributions:**

- The focus of the benchmark is novel. HARDMath2 targets the underrepresented area of applied mathematics requiring approximation techniques, a domain central to scientific and engineering reasoning yet largely absent from existing LLM evaluations. For example, asymptotics, WKB expansions, and nonlinear PDE analysis.

- An innovative problem curation process. The benchmark was created in a course-integrated framework, where students wrote original problems that had to defeat frontier LLMs, numerically verified their solutions in Colab notebooks, and peer-reviewed each other’s work.

- Clear presentation and reproducibility. The paper is clear, well-organized. The dataset is available to use on hugging face.

---

> ### Author Rebuttal · Authors · 2025-07-31
>
> Thank you very much for your feedback on our paper! We really appreciate your recognition of our benchmark’s contributions. Here are our responses and new results to incorporate your suggestions.
>
> **1. ```The current evaluation explicitly compares only the final numerical answer```**
>
> It is a good point that our method for evaluation only considers the final solution of the model’s responses. However, we deliberately chose this approach—as compared to using an LLM-as-a-judge with partial credit in our original HARDMath paper, the predecessor to this benchmark—since a direct numerical evaluation provides the most standardized and objective metric for the models’ performance, and is consistent with approaches taken in other recent mathematical benchmarks. Further, while in some applications having a chain of reasoning that results in an incorrect answer may still be useful for a user, there are many applications of LLMs where objective correctness is necessary, as the intervening reasoning may be screened or unavailable to an end user.
>
> However, we agree that this benchmark does have the potential to offer deeper insight into the model’s reasoning process in advanced mathematics. Students in the course, in addition to developing the benchmark problems, were asked to solve several of the problems as part of their final oral exam. We have used the detailed rubrics against which these students were scored to provide partial-credit scoring results using an LLM-judge. We provide one rubric (due to character limits) at the bottom of this response as well as partial-credit results for two reasoning models (Gemini 2.5 Pro and o4-mini) and the best non-reasoning model (DeepSeek V3).
>
> The results are here:
>
> | Model                      | Boundary Layers | Nonlinear PDE | WKB  |
> |---------------------------|------------------|----------------|------|
> | DeepSeek V3               | 57.6             | 55.3           | 55.3 |
> | o4-mini                   | 70.2             | 60.9           | 64.7 |
> | Gemini 2.5 Pro | 80.5             | 72.3           | 68.7 |
>
> We find that the partial credit scores are reasonably higher than our numerical evaluation scores. This can be explained by the fact that the top-performing models often understand the high-level approach to solving a problem but can struggle with getting the exact details correct. We also want to note that the LLM-judge scores are variable and model-dependent, whereas our original numerical evaluation is consistent. We used GPT-4o mini for the judge, but another model may produce very different results. It may also be challenging to calibrate the generosity of the judge, as some models may be more sycophantic than others [1]. However, we appreciate your feedback and are glad we were able to improve our benchmark results with these results.
>
> **2. Exclusion of some results**
>
> We agree that it is important to note which models struggled with instruction-following for transparency! In our evaluations, we found that DeepSeek R1 almost never provided the format we desired (a LaTeX \boxed{} solution), which is why we did not report results on it in our benchmark despite running full evaluations of the model. We similarly ran evaluations with Claude 4 Opus and Sonnet but did not report the scores due to issues with the maximum token parameter cutting off the models’ solution traces, which we were not able to resolve within Anthropic’s API. We experimented with various prompts, including ones that repeated the need for a LaTeX boxed solution for its final expression, but this did not resolve the issue either.
>
> In general, the models we included in the evaluation results rarely encountered such errors. We found that when they did yield parsing issues outside of the boxed, it was most usually due to the following reasons:
>
> - the model’s solution led to computational issues. For example, several solutions to the integral problems include expressions that never converged, meaning that the SymPy evaluator would crash or time-out. While this should be marked as incorrect, in this case we erred on the side of the model and skipped the question.
>
> - the model included Unicode, special characters, or additional json formatting in its solution. Our parser is already designed to replace or remove common Unicode/special characters/json commands with the correct LaTeX commands, but occasionally the model would write strange text that we could not replace, despite our system instructions asking for a LaTeX solution.
>
> - some models (particularly in the OpenAI family) seemed to have been overtrained on LaTeX formatting. They tended to insert a lot of extraneous formatting details, such as using “{+}” in place of “+” in their solutions to slightly adjust the spacing around the operators. While our parser could handle and remove most of these formatting issues, in some cases the models produced new obscure LaTeX formatting during evaluations which was missed by the parser.
>
> - the model wrote multiple expressions in the final solution box instead of one.
>
> We want to especially note that we do not include the problems which we are unable to parse in our evaluation results for fairness of the evaluations, instead of just scoring the model’s response as 0, which we believe would artificially deflate the scores. Additionally, our dataset included many more than the 211 problems reported in this paper. However, some of these problems consistently yielded parsing issues due to rare combinations of special characters or unusually complex combinations of LaTeX commands. Therefore, to increase the fairness and transparency of our benchmark’s evaluation reports, we did not include these problems in the dataset. For every problem in the dataset, at least one model could generate a properly formatted and parsable solution.
>
> We also believe that our prompts—which are detailed in Section A.2 of the appendix—are standard for benchmarks. We hope these additional details and results address your concerns; thanks again!
>
> __________
>
> References:
>
> [1] Sharma et al. (2024), Towards Understanding Sycophancy in Language Models

---

> ### Comment · Area_Chair_75KT · 2025-08-03
>
> Dear reviewer,
>
> Please read the rebuttal and provide your *final justification* and score.
>
> Best,
>
> AC

---

### Official Review · Reviewer_vdMP · 2025-07-03

**Rating:** 5
**Confidence:** 5

**Summary:**

This paper introduces HARDMath2, a novel benchmark for evaluating the advanced applied mathematics problem-solving capabilities of LLMs. The benchmark consists of 211 problems across six different problem types in applied mathematics (which are fundamental to applied science and engineering). These problems were built iteratively by the participants of a graduate class using LLMs to increase problem difficulty and also improve their own understanding.

The authors used the benchmark to evaluate a broad set of closed source and open source modes from the Gemini, GPT, OpenAI o-series models. The findings demonstrate that ‘thinking’ or ‘reasoning’ models perform better than non-thinking models. In general, models don’t do well on a majority of problem types highlighting gaps in multi step mathematical reasoning and problem solving in these models and also the utility of the proposed dataset for the same.

**Additional Feedback:**

1. In question 2 of the Paper checklist (on limitations), the justification given mentions that manual curation of problems is a limitation outlined in the conclusion section. The conclusion mentions automated generation is difficult due to generation-verification gap and would require more time and effort from the students (which means manual intervention is still needed). Also, on the other hand manual curation by participants of a graduate class serves as a novel pedagogy demonstrating the usage of AI in education (A contribution of the paper I suppose alongwith dataset). So what is mentioned as a limitation here, seems more like the central premise of this work. Could the authors clarify this?

2. The authors could have included a dedicated limitations or Impact statement in the main paper or appendix. I believe this will help further usage and research on this topic.

**Dataset Code Accessibility:**

Yes

**Dataset Code Comments:**

The entire code for the dataset evaluation is provided in a public github repository. The dataset is well structured, documented and is also made available through huggingface.

**Ethical Considerations:**

No, there are no or only very minor ethics concerns

**Final Justification:**

One of my major concerns was dilution of the evaluations. The authors have clarified this in detail and agreed to include the details on the failure modes of models, parsers etc. in the revised version of the paper.
Also, in response to a query by me on partial scoring to just understand the authors thoughts, the authors have not only clarified, but gone above and beyond and worked up a new LLM-as-ajudge eval for partial crediting during the rebuttal phase and it has become one of the contributions of the work. I would appreciate the authors for developing and including a new methodology in their work during the rebuttal phase. The work also is of the firsts in its kind and positions itself very clearly in the literature. I have increased my rating by a point.

**Limitations Weaknesses:**

The HARDMATH work used few-shot CoT prompting, does HARDMATH2 do that (Doesn't seem as couldn’t find the code for the same in the Github repository)? If yes, how much of a difference in performance does that make? If there are very similar examples or examples which have similar techniques/subroutines, does the LLM performance improve?
I believe it would be very interesting to see how the models behave when tested using manual crafting of a few-shot prompts or even by using RAG.

Is there a way for scoring intermediate or ‘reasoning’ steps in some form, as in the setup chosen clearly only the final answer is being evaluated?
It might be very difficult for the model to come up with the exact \LaTeX expression for the final answer (as per the dataset schema). Also since these problems require complex mathematical reasoning and decision making, understanding the intermediate steps becomes crucial to assess the models reasoning capabilities. I presume there would be some problems where it got the correct reasoning but arrived at an incorrect or partially correct solution.
Could LLM + Mathematical tool, LLM-as-a-judge, human manual grading or multiple choice have been used to evaluate? (I do not expect these to be part of this paper, I want to know the authors thoughts on how each method compares and justification for their dataset)

I am pleased to see the inclusion of the section A.4 in Appendix which highlights a major issue in evaluating the models. If the models fail to give response because of non-adherence to the format, it dilutes the metrics in some sense, if a model might be good at mathematical reasoning but is maybe incapable of adding the expression in the boxed environment in its response, then the performance measurements do not accurately capture the mathematical capabilities.  Can multiple/alternative response formats or evaluation of the entire stepwise solution be incorporated. Can the authors possibly add a few more detailed pointers on such weaknesses and challenges along with potential mitigation strategies for the same?

**Strengths Contributions:**

Alongwith contribution of a new dataset, the authors (and also the participants in the graduate class therein) provide a method to create similar such challenging datasets in other areas. Also the methodology serves as a good use case for the usage of AI in education and learning where students improve their understanding and to create and solve harder problems. The pedagogical framework can be applied to any quantitative field in science and engineering for designing difficult problems.

It has been noted that there is a gap in the datasets for advanced applied mathematics (which is also very useful as a subroutine in advanced physics problems). The dataset includes problems whose solutions require considering different regimes, choice of approximation methods, number of steps in expansions etc.
The dataset consists of 6 types of problems: nonlinear PDEs, nonlinear ODEs, WKB approximations, boundary layers and asymptotic series. This is fundamentally different from existing math datasets.

Another highlight of the work is the dedicated effort that has been invested to make the problems difficult to solve for the LLMs, by incorporating various handcrafted strategies for the same. (This is discussed in Section 4 of the main paper)

---

> ### Author Rebuttal · Authors · 2025-07-31
>
> Thank you for your thoughtful review! We have added several new results to incorporate your feedback, which we describe here.
>
> **Questions:**
>
> **1. ```The HARDMATH work used few-shot CoT prompting, does HARDMATH2 do that?```**
>
> We appreciate that you read the first HARDMath paper and looked at the code! We deliberately do not attempt few-shot prompting for a few reasons. First, recent reasoning models have found that few-shot prompting can hurt model performance. For example, the DeepSeek technical report [1] states that CoT in the prompt can degrade model performance, while OpenAI [2] recommends using zero-shot prompting before few-shot prompting. Additionally, mathematical benchmarks now emphasize direct evaluations without extra instructions or additional prompting strategies. For instance, the MathArena [3], FrontierMath [4], and Humanity’s Last Exam [5] benchmarks do not use CoT prompting.
>
> **2. ```Is there a way for scoring intermediate or ‘reasoning’ steps in some form?```**
>
> We focused on evaluating only the final solution of the model’s responses using our custom parser, as this most objectively checks the model’s performance. This direct comparison approach also aligns with those taken by other prominent mathematical benchmarks [3, 4], as objectively evaluating intermediate steps of a solution are challenging to scale. LLM-as-a-judge enables efficient grading but is not standardizable.
>
> However, your question is quite interesting in the context of this benchmark, which was developed as part of a graduate class. The students solved a subset of the benchmark problems themselves for a final oral examination. As such, we have used the detailed rubrics from grading the students in the course to provide partial-credit scoring using an LLM-judge. One sample rubric and the partial-credit results for the two best reasoning models (Gemini 2.5 Flash Thinking and o4-mini) and best non-reasoning model (DeepSeek V3) are provided below.
>
> The results are here:
>
> | Model                      | Boundary Layers | Nonlinear PDE | WKB  |
> |---------------------------|------------------|----------------|------|
> | DeepSeek V3               | 57.6             | 55.3           | 55.3 |
> | o4-mini                   | 70.2             | 60.9           | 64.7 |
> | Gemini 2.5 Pro | 80.5             | 72.3           | 68.7 |
>
> As expected, the partial-credit evaluations are higher than our original results. Using a detailed and custom-designed rubric for each problem type may be useful for reinforcement learning, where the reward captures the accuracy of the model’s reasoning process and not just the final solution. However, for the purposes of benchmarking, we still believe that a numerical check of the model’s final answer is the most reliable way of evaluating model capabilities, as there is no room for score inflation or inconsistency.
>
> **3. ```Could LLM + Mathematical tool, LLM-as-a-judge, human manual grading or multiple choice have been used to evaluate?```**
>
> We believe that if partial-credit scoring is desired, the LLM-as-a-judge is the best option. Multiple-choice allows the model to potentially guess correctly, which may give the wrong impression of the model’s skill. Human manual grading is the most reliable method but depends on experts and is not scalable. We believe that our custom numerical evaluation method is the most reliable and consistent form of evaluation, since LLM-as-a-judge results can vary based on the specific model chosen as the judge.
>
> **4. ```Can the authors possibly add a few more detailed pointers on such weaknesses and challenges?```**
>
> We want to thank you for your concern regarding dilution of the performance metrics. We were also concerned about this, given that it can be difficult in an automated process to tell if it is the fault of the model or the fault of our parser, neither of which are reflective of the model’s mathematical capabilities. We exclude any response from a model that cannot be parsed correctly, either due to the model’s lack of formatting or due to issues with the parser, from a model’s performance metrics. Thus, any performance metric is calculated based on problems that the model was able to solve and that we were able to successfully grade, ensuring that our results are indicative of mathematical reasoning and not ancillary grading-related issues. In practice, this resulted in a very small number of skipped questions for each model.
>
> In terms of challenges, we found that DeepSeek R1 was unable to follow the instructions to place its solutions in a \boxed{} environment. We found that other low-scoring models (e.g., Claude 3.7 Sonnet) frequently omitted important details from the solution. These examples include using ellipses (...) or unspecific variables to substitute important mathematical details in its reasoning process, and then never substituting the variables or terms back into their final expression.
>
> Other challenges with non-adherence to standards related to some model’s tendency to over-format responses. For example, OpenAI’s 4o model focused on manipulating display spacing, such as using “{-}” to slightly adjust spacing around operators. These display-only additions needed to be deleted on a case-by-case basis. Further, some models seemed to inject many unicode symbols into their responses which needed to be replaced with plain text for parsing. Finally, in a few cases (especially with respect to boundary layer problems) models would return solutions that were not purely analytic expressions but contained nested integrals. While our parser could handle these integrals, sometimes the integrals in the models’ responses would not converge numerically and so were skipped after timing out.
>
> **Additional feedback section:**
>
> 1. **Clarifying dataset creation:** We appreciate you pointing out this potential area of confusion and can see why it may have risen based on how we wrote the paper. To clarify: we agree with you that one of the main contributions of the paper is in the process of curating a benchmark through a course, with the accompanying pedagogical benefits for the students. That being said, it is still true that manually curating high-quality datasets is generally a process that is challenging to scale. Other benchmarks most often involve hiring experts to create new questions in answers; in mathematics, this means paying PhD students and professors to write new problems or original solutions.
>
> We are able to develop a useful educational framework for creating new data. Instead of asking students to solve already existing (and solved) problems, we task students with creating new problems and making them harder for current models. This allows us to “kill two birds with one stone” in that we provide a valuable pedagogical experience for students that removes the need for “additional work” as well as a repeatable framework for generating novel data.
>
> 2. **Adding a dedicated limitations or impact section:** we will certainly work on making the limitations and impact of our benchmark clearer and more detailed in our revised paper! In particular, we are strong proponents of our framework for simultaneous benchmark creation and educational instruction at the graduate-level in STEM. We also plan to add the additional details described above regarding common failure modes of models, parsing failures, and improvements we make to the parser in our final manuscript.
>
> We hope our new results and additional details answer your questions! Thanks again for reading our paper.
>
> _________
>
> References:
>
> [1] Guo et al. (2025), DeepSeek-R1: Incentivizing Reasoning Capability in LLMs via
> Reinforcement Learning
>
> [2] OpenAI (2025), Best practices for prompt engineering with the OpenAI API
>
> [3] Balunović et al. (2025), MathArena: Evaluating LLMs on Uncontaminated Math Competitions
>
> [4] Glazer et al. (2025), FrontierMath: A Benchmark for Evaluating Advanced Mathematical Reasoning in AI
>
> [5] Phan et al. (2025), Humanity's Last Exam
>
> ________
>
> ***Grading rubric for WKB problems (5 points)***
>
> 1. Recognition of WKB applicability
> - 1 point: Clearly recognizes that the equation is suitable for WKB (i.e., contains a small parameter multiplying the highest derivative or rapidly varying solution), and explains why WKB is appropriate (e.g., discusses scales, nature of coefficients, physical context).
> - 0 points: Fails to recognize the need for WKB, uses an inappropriate method, or provides weak justification for why WKB is needed.
>
> 2. Correct Statement of the WKB ansatz
> - 1 point: States the correct WKB ansatz (e.g., $y(x) \sim \exp\left[\frac{1}{\epsilon} S(x)\right]$ or similar), and includes all relevant assumptions (e.g., expansion of $S(x)$, small parameter $\epsilon \)$.
> - 0 points: Ansatz is missing or incorrect.
>
> 3. Substitution and Derivation of the WKB Equation
> - 1 point: Correctly substitutes the ansatz into the original equation, carefully computes all derivatives, and systematically derives the WKB equation with all steps shown.
> - 0 points: No meaningful substitution or substitution/derivation is incorrect.
>
> 4. Dominant Balance and Leading Order Equation
> - 1 point: Correctly identifies the dominant balance in the WKB equation (e.g., recognizes that the $(S’)^2$ term dominates for small $\epsilon$), writes the leading-order equation, and justifies neglecting smaller terms.
> - 0 points: Fails to identify dominant balance or applies an otherwise incorrect dominant balance.
>
> 5. Solution for Leading Order and Interpretation
> - 1 points: Correctly solves the leading order equation for $S(x)$, writes the leading order solution for $y(x)$, and interprets/expresses the result in the correct form, including proper handling of constants or boundary conditions if needed.
> - 0 points: Solution is missing, misinterpreted, or otherwise incorrect.”
>
> Note: you should only award a 5/5 if the model's final solution EXACTLY matches the ground-truth solution.
> __________

---

> ### Comment · Area_Chair_75KT · 2025-08-03
>
> Dear reviewer,
>
> Please read the rebuttal and provide your *final justification* and score.
>
> Best,
>
> AC

---

> ### Author Response · Authors · 2025-08-03
> **Following up on our rebuttal: additional comment on alternative methods for grading**
>
> Dear reviewer, we would like to offer one additional point regarding your thoughtful suggestion to consider alternative ways for grading model solutions. Human manual grading can indeed be valuable for LLM evals but is primarily needed when the solutions of a model cannot be systematically assessed. This is clearest in the case of proof-based mathematics, where benchmarks like MathArena or the recently released OpenAI and Google DeepMind results on the International Mathematical Olympiad (IMO) required human experts to grade each step of a proof, as the final "result" is irrelevant if the proof is logically incorrect. However, in our case, the correctness of a solution depends only on whether the numerical evaluation at a point in the domain matches that of the ground-truth solution. Therefore, the classes of problems in HARDMath2 are extremely well-suited for our numerical final-result parser.
>
> However, assigning partial credit can be valuable in the pedagogical context of our benchmark. In addition to our WKB approximation rubric above, we will also include the other detailed rubrics in the final manuscript of our paper!

---

> ### Author Response · Authors · 2025-08-06
> **Following up on our rebuttal: detailed rubric and increasing difficulty of problems**
>
> Dear reviewer, we wanted to follow up on your suggestions. We have added new partial-credit evaluation results for several problem classes in our benchmark and three of the best models, using an LLM-as-a-judge with a detailed rubric. We have included one rubric based on the actual criteria used in the course's final exam so that the automated partial-credit scoring aligns with the course instructors. We also provide more details surrounding the common failure modes of models.
>
> Finally, we want to give some additional information below on the automatic component of our problem generation framework to help clarify how we systematically scale the difficulty of our problems.
> _________
>
> When we have an "easy" problem that a weaker model like Gemini 2.0 can already solve, we can use the following framework to systematically increasing the difficulty of that problem. Our framework relies on two instances of a powerful model (Gemini 2.5 Pro): one instance acts as both an all-knowing verifier and obfuscator, which has access to the original problem, the original solution, and all modifications to the original problem statement, while the second instance acts as a solver to test the difficulty of the new problem statement. The verifier first modifies the problem and passes it to the (context-less) solver, which attempts to solve the modified problem. The verifier, which knows the modification that was made, will check the solution generated by the solver and assess its correctness. If it deems the solution to be correct, it will again modify the problem statement, but if it finds the solution incorrect, we end the process there, generate a solution from the verifier, and manually check the solution provided by the model. Students found that the verifier was typically quite skilled at identifying flaws in the solver’s proposed solution.
>
> We use two classes of techniques that either leave the original solution unchanged or modify the solution. Methods to retain the original solution include the addition of cancellable terms, replacement of simple terms with other terms that are equivalent but algebraically more complex, and multiplication by a complex function that evaluates to 1 in the limit of interest. These do not require the verifier model above, as we can be certain that the original solution does not change and just check whether the solver’s solution matches the original solution. Alternatively, we can change the solution of the problem itself by adding or composing trigonometric, logarithmic, hyperbolic, or other special functions in the problem (e.g., $f(x)$ becomes $\sin(f(x))$. These problems are typically more difficult for the models to solve, and we require the human-in-the-loop to resolve the new problem to guarantee the solution’s correctness.
> ___________
> We hope this addresses your concerns! Please let us know if you have any additional questions.

---

> ### Author Response · Authors · 2025-08-08
> **Final summary of all additions and clarifications**
>
> Dear reviewer,
>
> In case it is helpful for your final assessment, here is a concise list of the improvements we have made following your suggestions.
>
> - **Partial-credit scoring:** we implemented an LLM-as-a-judge on the full solution trace using detailed, specific rubrics for several problem types from the course to assess final exams. We report the partial-credit scores and find an expected increase in the average score but note that the justifications provided by the LLM are sensible. We plan to provide an analysis of these reports in the Appendix due to limited space here.
>
> - **Clarified methodology:** we explain why we avoid using CoT and few-shot prompting in our evaluations, following the best practices provided by OpenAI and DeepSeek, and why final numerical answer checking as implemented by our custom parser is most effective for mathematical benchmarks.
>
> - **Failure modes:** we detailed the common failure modes of the LLMs in their responses and the mitigation strategies we implemented in our parser.
>
> - **Systematic problem modifier:** we describe the LLM-based solver-verifier framework for systematically increasing problem difficulty.
>
> We hope these updates resolve your questions!

---

> > ### Comment · Reviewer_vdMP · 2025-08-09
> >
> > Thanks for the response. I am particularly pleased to see the inclusion of LLM-as-a-judge eval based method for partial credit scoring of models and also of the solver-verifier framework. Thanks for the clarification on CoT prompting, evaluation techniques and on parsers etc. All of my questions and doubts have been very well answered! I have adjusted my rating accordingly.

---

> > > ### Author Response · Authors · 2025-08-09
> > >
> > > Thank you very much! We appreciate your feedback.

---

### Note · Authors · 2025-08-14

Dear reviewers and AC, thank you for reading our paper and for providing valuable suggestions. We are glad that our additional results and clarifications have proven useful. We believe our dataset contains a large set of unique problems compared to other widely-used benchmarks. We constructed the dataset differently from the previous iteration to increase the diversity and difficulty of problems; while this meant that we could not arbitrarily scale its size to thousands of problems, its quality is consequently much higher while still being larger than Olympiad exams and comparable to FrontierMath, which required months of contracting from professional mathematicians.

---

### Decision · Program_Chairs · 2025-09-18

**Decision:**

Accept (poster)

**Comment:**

This paper introduces HARDMath2, a benchmark for evaluating the advanced applied mathematics problem-solving capabilities of LLMs. The benchmark consists of 211 problems across six different problem types in applied mathematics (which are fundamental to applied science and engineering). These problems were built iteratively by the participants of a graduate class using LLMs to increase problem difficulty and also improve their own understanding.

The authors used the benchmark to evaluate closed source and open source modes from the Gemini, GPT, OpenAI o-series models. The findings demonstrate that ‘thinking’ or ‘reasoning’ models perform better than non-thinking models. In general, models don’t do well on a majority of problem types highlighting gaps in multi-step mathematical reasoning and problem-solving in these models.

After the rebuttal, the paper receives three Accept and one Borderline Accpet. Majority of the concerns were solved. Reviewer 9dGN still has doubts on dataset scale. The AC encourages the authors to further enlarge the dataset in terms of both scales and domain coverage as much as possible in the final version.